# STABLE SEGMENT ANYTHING MODEL

**Qi Fan[1], Xin Tao[2,*], Lei Ke[3], Mingqiao Ye[4], Pengfei Wan[2], Di Zhang[2],**
**Yu-Wing Tai[5], Chi-Keung Tang[6]**
[1] Nanjing University, [2] Kuaishou Technology, [3] Carnegie Mellon University,
[4] EPFL, [5] Dartmouth College, [6] The Hong Kong University of Science and Technology

## ABSTRACT

The Segment Anything Model (SAM) achieves remarkable promptable segmentation given high-quality prompts which, however, often require good skills to specify. To make SAM robust to casual prompts, this paper presents the first comprehensive analysis on SAM's segmentation stability across a diverse spectrum of prompt qualities, specifically imprecise bounding boxes and insufficient points. Our key finding reveals that given such low-quality prompts, SAM's mask decoder tends to activate image features that are biased towards the background, or confined to specific object parts. To mitigate these issues, our solution consists of calibrating solely SAM's mask attention by adjusting the sampling locations and amplitudes of image features, while the original SAM model architecture and weights remain unchanged. Consequently, our deformable sampling plugin (DSP) enables SAM to adaptively shift attention to the prompted target regions in a data-driven manner. During inference, dynamic routing plugin (DRP) is proposed that toggles SAM between the deformable and regular grid sampling modes, conditioned on the input prompt quality. Thus, our solution, termed Stable-SAM, offers several advantages: 1) improved SAM's segmentation stability across a wide range of prompt qualities, while 2) retaining SAM's powerful promptable segmentation efficiency and generality, with 3) minimal learnable parameters (0.08 M) and fast adaptation. Extensive experiments validate the effectiveness and advantages of our approach, underscoring Stable-SAM as a more robust solution for segmenting anything. Codes are at https://github.com/fanq15/Stable-SAM.

## 1 INTRODUCTION

The recent Segment Anything Model (SAM (Kirillov et al., 2023)) stands a significant milestone in image segmentation, attributed to its superior zero-shot generalization ability on new tasks and data distributions. Empowered by the billion-scale training masks and the promptable model design, SAM generalizes to various visual structures in diverse scenarios through flexible prompts, such as box, point, mask or text prompts. Facilitated by high-quality prompts, SAM has produced significant performance benefit for various important applications, such as healthcare (Huang et al., 2023b; Mazurowski et al., 2023), remote sensing (Wen et al., 2023; Ding et al., 2023), self-driving (Dikshit et al., 2023; Fan et al., 2022b), agriculture (Nguyen et al., 2023; Liu, 2023), *etc*.

Previous works mainly focus on improving SAM's segmentation performance assuming high-quality prompts are available, such as a tight bounding box (*e.g.*, produced by SOTA detectors (Jia et al., 2023; Zhang et al., 2023a; Yang et al., 2022)) or sufficient points (Ke et al., 2023) (*e.g.*, 10 points) for the target object. However, in practice SAM and interactive segmentation are often given inaccurate or insufficient prompts casually marked up by users as inaccurate box, or very sparse points are given, especially in the crowd-sourcing annotation platform. Such inaccurate prompts often mislead SAM to produce unstable segmentation results as shown in Figure 1. Unfortunately, however, this critical issue has been largely overlooked, even though the suboptimal prompts and the resulting segmentation stability problem are quite prevalent in practice .

Note that there is no proper off-the-shelf solution for solving SAM's segmentation stability problem with inaccurate prompts. Simply finetuning SAM's mask decoder with imprecise prompts may easily

---

*Corresponding Author: Xin Tao

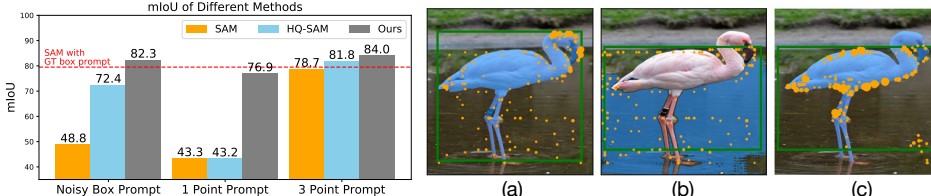

Figure 1: To illustrate SAM's instability, the left figure compares the performance among SAM, HQ-SAM and our Stable-SAM, when provided with suboptimal prompts. Our Stable-SAM consistently surpasses other methods across prompts of different quality, demonstrating better or comparable performance to the SAM prompted by ground truth box. The right figure displays the predicted masks and sampled important image features of SAM and Stable-SAM prompted by the bounding box (in green color), with larger orange circles indicating higher attention weights. (a) SAM yields satisfactory segmentation results when provided with a high-quality box prompt. (b) SAM can be very unstable, as shown here even a minor prompt modification makes SAM segment the background instead. SAM incorrectly segments the background, where the inaccurate box prompt misleads SAM to spend more attention to the background. (c) Our Stable-SAM accurately segments the target object by shifting more feature sampling attention to it.

lead to catastrophic forgetting, undermining the integrity of the highly-optimized SAM model and thus sacrificing the zero-shot segmentation generality. Although in the image domain deformable attention (Dai et al., 2017) has shown impressive efficacy on adaptively shifting the model attention to informative regions, which may naturally address the attention drift issue caused by the misleading prompts, a straightforward implementation of this idea can again compromise SAM's integrity.

In this paper we present the first comprehensive analysis on SAM's segmentation stability across a wide range of prompt qualities, with a particular focus on low-quality prompts such as imprecise bounding boxes or points. Our findings demonstrate that, when fed with imprecise prompts, the SAM's mask decoder is likely to be misguided to focus on the background or specific object parts, where the cross-attention module is inclined to aggregate and activate image features of these regions when mutually updating the prompt and image tokens. Such collaborative token updating mechanism usually suffers from attention drift, where the suboptimal prompts misleadingly shift attention from the target object to the background areas or specific object parts. The attention drift is accumulated and propagated from the suboptimal prompt to the unsatisfactory segmentation results.

To address this issue, we present a novel deformable sampling plugin (DSP) with *two* key designs to improve SAM's stability while maintaining its zero-shot generality. Our key idea is to adaptively calibrate SAM's mask attention by adjusting the attention sampling positions and amplitudes, while keeping the original SAM model unchanged: *1)* we employ a small offset network to predict the corresponding offsets and feature amplitudes for each image feature sampling locations, which are learned from the input image feature map; *2)* then, we adjust the feature attention by resampling the deformable image features at the updated sampling locations of the cross-attention module in SAM's mask decoder, keeping the original SAM model unchanged. In doing so, we can shift the feature sampling attention toward informative regions which is more likely to contain target objects, and meanwhile avoiding the potential model disruption of the original highly-optimized SAM. Finally, to effectively handle both the high- and low-quality prompts, we propose a dynamic routing module to toggle SAM between deformable and regular grid sampling modes. A simple and effective robust training strategy is proposed to facilitate our Stable-SAM to adapt to prompts of diverse qualities.

Thus, our method is unique in its idea and design on solely adjusting the feature attention without involving the original model parameters. In contrast, conventional deformable attention methods (Dai et al., 2017; Xia et al., 2022) update the original network parameters, which is undesirable when adapting powerful foundation models involving finetuning such large models in data-scarce scenarios.

Our model, Stable-SAM, benefits both the selective deformable attention and the powerful original SAM model, with minimal addition of computational overhead and parameters. First, the SAM's segmentation stability is substantially improved across a wide range of prompt qualities, especially with low-quality prompts. Besides, the original SAM's powerful promptable segmentation efficiency and generality are preserved well even in the data-scarce scenarios. Extensive experiments across multiple datasets validate the effectiveness and advantages of our approach, underscoring its potential as a robust solution for segmentation tasks.

## 2 RELATED WORKS

**Segment Anything Model.** The recent Segment Anything Model (Kirillov et al., 2023; Ravi et al., 2024) has gained widespread recognition, attributed to its remarkable performance and generalization in image segmentation. SAM has been applied in a wide range of downstream tasks and applications, including medical images (Ma et al., 2024; Zhang & Liu, 2023; Liu et al., 2024; Leng et al., 2024), object tracking (Zou et al., 2024), data annotation (He et al., 2023; Wang et al., 2024), 3D reconstruction (Cen et al., 2024; Yin et al., 2023), robotics (Huang et al., 2023a), and multimodal tasks (Mo & Tian, 2023; Zhang et al., 2023b; Wang et al., 2023). Some researchers attempt to address SAM's computational limitations and improve its efficiency. Some works (Zhang et al., 2023c; Liu et al., 2023) focus on improving SAM's segmentation quality and generalization to downstream applications. Thus the foundation model finetuning methods (Chen et al., 2023; Wu et al., 2023) are widely adopted for fast and effective SAM adaptation in specific segmentation scenarios.

**Improving Segmentation Quality.** Researchers have proposed various methods to enhance the quality and accuracy of semantic segmentation methods. Early methods incorporate graphical models such as CRF (Krähenbühl & Koltun, 2011) or region growing (Dias & Medeiros, 2019) as an additional post-processing stage, which are usually training-free. Many learning-based methods design new operators (Ke et al., 2022a;b; Kirillov et al., 2020) or utilize additional refinement stage (Cheng et al., 2020; Shen et al., 2022). Recently, methods such as Mask2Former (Cheng et al., 2022) and SAM (Kirillov et al., 2023) have been introduced, which address open-world segmentation by introducing prompt-based approaches. Along this line, a series of improvements (Ke et al., 2023; Li et al., 2023) have been proposed, focusing on prompt-tuning and improving the accuracy of segmentation decoders. However, these methods overlook a crucial aspect, which is how to generate high-quality segmentation results in cases where the prompt is inaccurate. This is precisely the problem that our method aims to address.

**Tuning Foundation Models.** Pretrained models have played an important role since the very beginning of deep learning (Krizhevsky et al., 2012; Simonyan & Zisserman, 2015; He et al., 2016). Despite zero-shot generalization grows popular in foundation models of computer vision and natural language processing (Bommasani et al., 2021; Brown et al., 2020), tuning methods such as adapter (Hu et al., 2022) and prompt-based learning (Houlsby et al., 2019; Hu et al., 2022) have been proposed to generalize these models to downstream tasks (Fan et al., 2020; 2022a). These methods typically involves additional training parameters and time. We propose a new method that makes better use of existing features with minimal additional methods and can also produce competitive results.

**Deformable Attention.** Deformable convolution (Dai et al., 2017; Zhu et al., 2019) has been proved effective to help neural features attend to important spatial locations. Recently, it has also been extended to transformer-based networks (Chen et al., 2021; Yue et al., 2021; Zhu et al., 2020; Xia et al., 2022). Such deformed spatial tokens are especially suitable for our task, which requires dynamically attending to correct regions given inaccurate prompts. However, previous deformable layers involve both offset learning and feature learning after deformation. In this paper, we propose a new approach to adjust the feature attention by simply sampling and modulating the features using deformable operations, without the need to train subsequent layers.

## 3 SAM STABILITY ANLAYSIS

We perform empirical studies to illustrate the segmentation instability of the current SAM with prompts of differing quality, thereby justifying our Stable SAM approach.

Prior segmentation studies have focused on achieving high prediction accuracy, gauged by the Intersection-over-Union (IoU) between the predicted and ground truth masks. This focus on high performance is justified as segmentation models typically produce deterministic masks for given input images, without requiring additional inputs. However, SAM's segmentation output depends on both the image and the prompts, with the latter often varying in quality due to different manual or automatic prompt generators. In practical applications of SAM, segmentation targets are typically clear and unambiguous, independent of prompt quality.

**Segmentation Stability Metric.** Motivated by this application requirement, we introduce the segmentation stability metric. Specifically, SAM is capable of producing a set of binary segmentation maps $M \in \mathcal{R}^{B \times H \times W}$ for a single target object using $B$ prompts of differing qualities. We define the

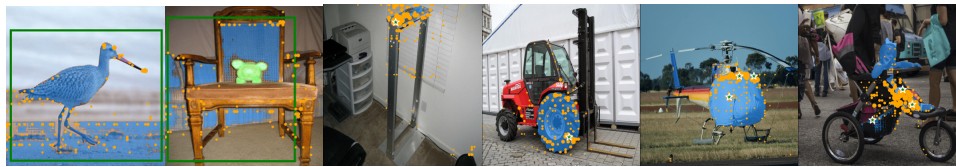

Figure 2: SAM performs badly when dealing with suboptimal prompts. This is mainly caused by the undesirable feature attention, focusing on the background or specific object parts. The important features are highlighted by the orange circles, with larger radius indicating higher attention score. The green boxes denote the box prompts with added noise to the groundtruth boxes. The stars denote the point prompts which are randomly sampled from the groundtruth masks.

segmentation stability (ST) within the set as:

$$ST = \frac{1}{B} \sum_{i=1}^{B} \text{IoU}(M_i, M_{\text{union}}), \tag{1}$$

where $\text{IoU}(M_i, M_{\text{union}})$ represents the Intersection-over-Union between the $i$-th segmentation map $M_i$ and the collective foreground region $\bigcup_i^B M_i$ of all maps. This new metric assesses the consistency across segmentations in each prediction, serving as a reliable indicator of stability, even without access to the ground truth masks.

**Model and Evaluation Details.** The released SAM is trained with crafted prompts on large-scale SA-1B dataset. We evaluate the segmentation accuracy and stability of the ViT-Large based SAM with different prompt types and qualities, including box prompts with added noise (we insert the uniform noise with the noise scale 0.4 into the box height, width and center position) and point prompts with varying numbers of points (1, 3, 5, 10 positive points randomly selected from the ground truth mask). For every input image and prompt type, we randomly select 20 prompts to compute their segmentation stability, average mask mIoU, and boundary mBIoU scores. The evaluation utilizes four segmentation datasets as in HQ-SAM: DIS (Qin et al., 2022) (validation set), ThinObject-5K (Liew et al., 2021) (test set), COIFT (Mansilla & Miranda, 2019), and HR-SOD (Zeng et al., 2019).

Table 1 tabulates that SAM's segmentation accuracy and stability significantly decrease with low-quality prompts, such as imprecise box prompts or point prompts with minimal points. The varying segmentation accuracy and stability indicates that SAM's mask decoder performs distinctly when dealing with prompts of varying qualities.

Table 1: SAM's segmentation accuracy and stability under prompts of varying quality. All evaluation metrics are averaged on four HQ datasets.

| Metric | GT Box | Noisy Box | 1 Point | 3 Points | 5 Points | 10 Points |
|---|---|---|---|---|---|---|
| mIoU | 79.5 | 48.8 | 43.3 | 78.7 | 83.3 | 84.8 |
| mBIoU | 71.1 | 42.1 | 37.4 | 69.5 | 74.2 | 76.0 |
| ST | - | 39.5 | 45.1 | 79.3 | 84.7 | 87.5 |

We visualize the image activation map for the token-to-image cross-attention in SAM's second mask decoder layer to better understand its response to low-quality prompts. We focus on the second mask decoder layer for visualization because its cross-attention is more representative, benefiting from the input tokens and image embedding collaboratively updated by the first mask decoder layer. Figure 2 demonstrates that an inaccurate box prompt causes SAM's mask decoder to miss regions of the target object while incorrectly incorporating features from the background, or focusing on specific object parts. It consequently leads to degraded segmentation accuracy and stability.

Overall, the above empirical evidence suggests that SAM potentially suffers from the attention drift issue, where suboptimal prompts misleadingly shift attention from the target object to background areas or specific object parts, thereby compromising the segmentation accuracy and stability. This motivates us to calibrate SAM's mask attention by leveraging learnable offsets to adjust the attention sampling position towards the target object regions, thus boosting segmentation accuracy and stability.

## 4    STABLE SEGMENT ANYTHING MODEL

We first revisit the recent Segment Anything Model (SAM) and deformable attention mechanism.

**Segment Anything Model.** SAM (Kirillov et al., 2023) is a powerful promptable segmentation model. It comprises an image encoder for computing image embeddings, a prompt encoder for embedding

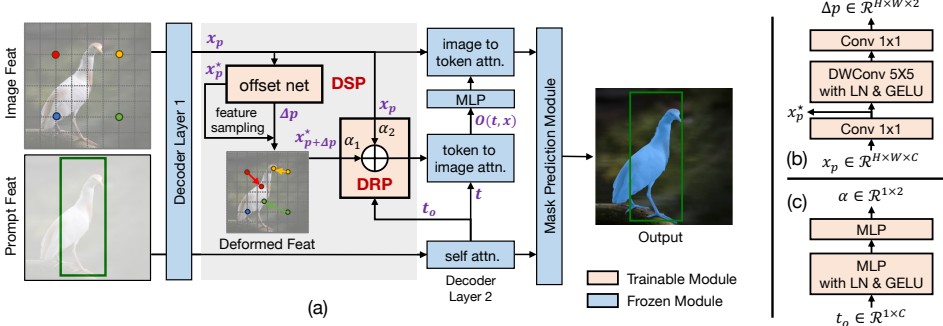

Figure 3: (a) An illustration of our deformable sampling plugin (DSP) and deformable routing plugin (DRP) in SAM's mask decoder transformer. DSP employs a small (b) offset network to predict the feature sampling offsets and amplitudes. Subsequently, DSP calibrates the feature attention by resampling deformable image features at the updated sampling locations, and feeds them into SAM's token-to-image attention. DRP employs a small (c) MLP network to regulate the degree of DSP activation based on the input prompt quality. Note that our DSP adaptively calibrates solely SAM's mask attention without altering the original SAM model.

prompts, and a lightweight mask decoder for predicting segmentation masks by combining the two information sources. The fast mask mask decoder is a two-layer transformer-based decoder to collaboratively update both the image embedding and prompt tokens via cross-attention. SAM is trained on the large-scale SA-1B dataset.

**Deformable Attention.** Deformable attention (Xia et al., 2022) is a mechanism that enables the model to focus on a subset of key sampling points instead of the entire feature space. This mechanism naturally addresses the attention shift problem in SAM caused by low-quality prompts.

In the standard self-attention, given a feature map $x \in \mathcal{R}^{H \times W \times C}$, the attention weights are computed across all spatial locations within the feature map.

In the deformable attention (Xia et al., 2022), a uniform grid of points $r \in \mathcal{R}^{H_G \times W_G \times 2}$ are first generated as the references[1] with the sampled image feature $x_r \in \mathcal{R}^{H_G \times W_G \times C}$. Subsequently, a convolutional offset network $\theta_{\text{offset}}$ predicts the offset $\Delta r = \theta_{\text{offset}}(x_r)$ for each reference point. The new feature sampling locations are given by $r + \Delta r \in \mathcal{R}^{H_G \times W_G \times 2}$. The resampled deformable image features $x_{r+\Delta r} \in \mathcal{R}^{H_G \times W_G \times C}$ are then utilized as the features in the attention module.

Note that conventional deformable attention optimizes both the offset network and attention module. Thus directly applying deformable attention to SAM is usually suboptimal, because altering SAM's original network or weights, *e.g.*, substituting SAM's standard attention with deformable attention and retraining, may compromise its integrity.

## 4.1 DEFORMABLE SAMPLING PLUGIN

To address the attention drift issue while preserving SAM's integrity, we propose a novel deformable sampling plugin (DSP) module on top of SAM's original token-to-image cross-attention module, as shown in Figure 3.

Specifically, given the prompt token feature $t \in \mathcal{R}^{\text{T} \times C}$ and image feature $x_p \in \mathcal{R}^{H \times W \times C}$, the token-to-image cross-attention is:

$$\text{CAttn}(t, x) = \sigma(Q(t) \cdot K(x_p)^T) \cdot V(x_p), \tag{2}$$

where $p \in \mathcal{R}^{H \times W \times 2}$ represents the image feature spatial sampling locations, $\sigma$ denotes the softmax function, and $Q, K, V$ are the query, key, and value embedding projection functions, respectively.

Our DSP adaptively calibrate the feature attention by adjusting solely feature sampling locations and amplitudes without altering the original SAM model. Specifically, we utilize an offset network $\theta_{\text{offset}}$ to predict the feature sampling offset $\Delta p \in \mathcal{R}^{H \times W \times 2}$, akin to that in deformable attention:

$$\Delta p = \theta_s(\theta_{\text{offset}}(x_p)), \tag{3}$$

---

[1]With the grid size downsampled from the input feature map spatial size (H, W) by a factor of $s$, thus $H_G = H/s$ and $W_G = W/s$.

where $\theta_s$ is a scale function $s_p \cdot \tanh(*)$ to prevent too large offset, and $s_p$ is a pre-defined scale factor. The offset network $\theta_{\text{offset}}$ consists of a $1 \times 1$ convolution, a $5 \times 5$ depthwise convolution with the layer normalization and GELU activation, and a $1 \times 1$ convolution. The updated feature sampling locations are $p + \Delta p$. The numerical range of both $p$ and $p + \Delta p$ is clamped in $\{(0,0), ..., (H-1, W-1)\}$, which is then normalized to the range $[-1, 1]$ for feature sampling. The feature amplitudes are predicted by the first convolutional layer and the image features $x_p$ are thus updated as $x_p^\star$, which are used solely for computing the feature attention.

Subsequently, we resample and modulate deformable image features $x_{p+\Delta p}^\star \in \mathcal{R}^{H \times W \times C}$ at the updated sampling locations $p + \Delta p$ with the learned feature amplitudes for keys and values. Thus, our DSP calibrates the token-to-image cross-attention of SAM's mask decoder as:

$$\text{DCAttn}(t, x) = \sigma(Q(t) \cdot K(x_{p+\Delta p}^\star)^T) \cdot V(x_{p+\Delta p}^\star). \tag{4}$$

As $p + \Delta p$ is fractional, we apply a bilinear interpolation to compute $x_{p+\Delta p}^\star$ as in Deformable DETR (Zhu et al., 2020).

Note that our DSP only trains the deformable offset network to predict new feature sampling locations $p + \Delta p$ and feature amplitudes, and feeds the resampled and modulated deformable features $x_{p+\Delta p}^\star$ to SAM's cross-attention module. Thus, the original SAM model remains unchanged.

## 4.2 DYNAMIC ROUTING PLUGIN

While our DSP can effectively handle suboptimal and even erroneous prompts, by redirecting SAM's attention to informative regions which are more likely to contain the target objects, high-quality prompts can typically direct the model's attention correctly to target regions. Thus, it is essential to properly control the DSP's activation to prevent unwanted attention shifts.

To address this issue, we propose a novel dynamic routing plugin (DRP) that regulates the degree of DSP activation based on the input prompt quality. The DRP can be formulated as follows:

$$\alpha = \sigma(\text{MLP}(t_o)) \cdot s, \tag{5}$$

where $t_o \in \mathbb{R}^{1 \times C}$ is the prompt token feature corresponding to the output mask, $\text{MLP}$ refers to a small MLP network that includes an MLP layer with LayerNorm and GELU activation, as well as an output MLP layer; $s$ denotes a learnable scale and $\sigma$ denotes the softmax function.

We utilize the predicted values of $\alpha = [\alpha_1, \alpha_2] \in \mathbb{R}^{1 \times 2}$ to adaptively route SAM between DSP and original SAM's attention mechanism. Consequently, the token-to-image cross-attention output $\text{O}(t, x)$ can be formulated as:

$$\text{O}(t, x) = \text{CAttn}(t, \alpha_1 \cdot x_{p+\Delta p}^\star + \alpha_2 \cdot x_p) \tag{6}$$

This soft dynamic routing strategy allows SAM to benefit from both DSP and its original zero-shot generality, contingent upon the quality of the prompt.

## 4.3 ROBUST TRAINING STRATEGY

We propose a simple and effective robust training strategy (RTS) to assist our model to learn how to correct SAM's attention when adversely affected by bad prompts.

**Robust Training Against Inaccurate Prompts.** SAM's training, including HQ-SAM (Ke et al., 2023), typically utilizes high-quality prompts given by precise bounding boxes or multiple points to accurately identify the target object. To address inaccurate prompts, our RTS incorporates prompts of varying qualities during training. These prompts include groundtruth boxes, box prompts with added noise (noise scale 0.4), and point prompts with varying numbers of points (1, 3, 10 positive points randomly chosen from the ground truth mask).

**Robust Training Against Ambiguous Prompts.** In real segmentation scenarios, target objects often occur in cluttered environment, either occluding others or being occluded. Even given an accurate, tight bounding box, objects other than the target object will be enclosed. On the other hand, target objects are typically unambiguous even other objects are enclosed. For instance, in MS COCO, beds (occluded by quilt) are consistently regarded as target objects; the model must accurately segment the entire bed including accessories such as pillows and bedding. Thus, SAM's original

Table 2: Method comparison on four HQ datasets under prompts of varying quality. All models (except for methods in the first group) are trained on HQSeg-44K dataset.

| Model | Epoch | Noisy Box | | | 1 Point | | | 3 Points | | |
|---|---|---|---|---|---|---|---|---|---|---|
| | | mIoU | mBIoU | ST | mIoU | mBIoU | ST | mIoU | mBIoU | ST |
| SAM (baseline) | - | 48.8 | 42.1 | 39.5 | 43.3 | 37.4 | 45.1 | 78.7 | 69.5 | 79.3 |
| PA-SAM (Xie et al., 2024) | - | 51.2 | 44.4 | 41.8 | 45.3 | 39.5 | 47.2 | 79.6 | 70.1 | 80.0 |
| CAT-SAM (Xiao et al., 2024) | - | 51.5 | 44.8 | 42.1 | 45.7 | 39.9 | 47.6 | 80.0 | 70.6 | 80.5 |
| RobustSAM (Chen et al., 2024) | - | 51.7 | 44.9 | 42.3 | 45.9 | 40.2 | 47.7 | 80.4 | 71.1 | 81.0 |
| SAM 2 (Ravi et al., 2024) | - | 52.4 | 45.3 | 43.1 | 46.7 | 41.1 | 48.5 | 81.1 | 71.8 | 81.7 |
| FT-SAM (finetuning SAM's whole model) | 12 | 32.5 | 27.7 | 24.1 | 28.6 | 22.8 | 30.3 | 46.2 | 35.4 | 43.1 |
| DT-SAM (finetuning SAM's mask decoder) | 12 | 70.6 | 60.4 | 64.0 | 43.1 | 43.2 | 37.9 | 80.3 | 71.6 | 80.5 |
| PT-SAM (finetuning SAM's prompt token) | 12 | 70.8 | 60.2 | 64.1 | 43.0 | 42.9 | 38.3 | 80.1 | 71.8 | 80.4 |
| SAM with LoRA (Hu et al., 2022) | 12 | 70.3 | 60.6 | 63.7 | 42.3 | 43.2 | 37.2 | 79.5 | 71.2 | 79.6 |
| SAM with Adapter (Chen et al., 2022) | 12 | 70.5 | 60.0 | 63.2 | 42.7 | 43.3 | 37.5 | 79.8 | 71.4 | 80.0 |
| HQ-SAM (Ke et al., 2023) | 12 | 72.4 | 62.8 | 65.5 | 43.2 | 44.6 | 37.4 | 81.8 | 73.7 | 81.4 |
| Stable-SAM | 1 | 82.3 | 74.1 | 82.3 | 76.9 | 68.4 | 71.1 | 84.0 | 75.8 | 84.9 |
| Stable-SAM 2 | 1 | **83.5** | **75.3** | **83.4** | **78.0** | **69.6** | **72.2** | **85.1** | **76.9** | **86.0** |

ambiguity-aware solution, which predicts *multiple* masks for a single prompt, is generally suboptimal in well-defined realistic applications. To address such "ambiguous" prompts, our RTS incorporates synthetic occlusion images to make SAM conducive to accurately segment target objects. We include the implementation details of the occlusion image synthesis in the supplementary materials.

Our RTS is general and applicable to various SAM variants to improve their segmentation stability. Notably, our Stable-SAM with DSP and DRP experience the most substantial improvements from the application of RTS.

## 5 EXPERIMENTS

**Datasets.** For fair comparison we keep our training and testing datasets same as HQ-SAM (Ke et al., 2023). Specifically, we train all models on HQSeg-44K dataset, and evaluate their performance on four fine-grained segmentation datasets, including DIS (Qin et al., 2022) (validation set), ThinObject-5K (Liew et al., 2021) (test set), COIFT (Mansilla & Miranda, 2019) and HR-SOD (Zeng et al., 2019). Furthermore, we validate the model's zero-shot generalization ability on three challenging segmentation benchmarks, including COCO (Lin et al., 2014), SGinW (Zou et al., 2023) and MESS (Blumenstiel et al., 2023). SGinW contains 25 zero-shot in-the-wild segmentation datasets. MESS is a large-scale benchmark for holistically evaluating the zero-shot segmentation performance.

**Input Prompts.** We evaluate model's accuracy and stability with prompts of differing type and quality, as described in Sec. 3. For MS COCO and SGinW, we do not use the boxes generated by SOTA detectors (Zhang et al., 2023a; Jia et al., 2023) as the box prompt. This is because their predicted boxes are typically of high quality and cannot effectively evaluate the model's segmentation stability in the presence of inaccurate boxes. Instead, we introduce random scale noises into the ground truth boxes to generate noisy boxes as the prompts. Specifically, to simulate inaccurate boxes while still having some overlap with the target object, we select noisy boxes that partially overlap with the ground truth boxes with IoU ranges of 0.5–0.6 and 0.6–0.7. We also evaluate our method using the box prompts generated by SOTA detectors.

### 5.1 COMPARISON WITH SAM VARIANTS

We compare our method with SAM and three powerful SAM variants. HQ-SAM is a recent powerful SAM variant for producing high-quality masks. We also try two popular model finetuning methods, LoRA (Hu et al., 2022) and Adapter (Chen et al., 2022), and three simple SAM variants by finetuning the SAM's whole model, its mask decoder and the prompt token, *i.e.*, FT-SAM, DT-SAM and PT-SAM, respectively. All our Stable-SAM models are trained by just one epoch for fast adaptation unless otherwise stated. All other models are trained 12 epochs. More experimental results and implementation details are included in the supplementary material.

**Stability Comparison on Four HQ Datasets.** Table 2 shows the segmentation accuracy and stability on four HQ datasets, when models are fed with suboptimal prompts. Notably, the use of noisy box prompts significantly reduces SAM's performance, as evidenced by the drop from 79.5/71.1 (as

Table 3: Comparison on MS COCO and SGinW datasets. All models (except for the SAM baseline) are trained on HQSeg-44K dataset. All models are prompted by noisy boxes (N-Box) that overlap with the ground truth boxes, with IoU ranges of 0.5-0.6 and 0.6-0.7.

| Model | Epoch | MS COCO N-Box (0.5-0.6) mAP | mAP$_{50}$ | N-Box (0.6-0.7) mAP | mAP$_{50}$ | SGinW N-Box (0.5-0.6) mAP | mAP$_{50}$ | N-Box (0.6-0.7) mAP | mAP$_{50}$ | Learnable Params | Mem. | FPS |
|---|---|---|---|---|---|---|---|---|---|---|---|---|
| SAM (baseline) | - | 27.3 | 60.2 | 40.9 | 75.0 | 26.0 | 60.8 | 39.5 | 73.2 | (1191 M) | 7.6 G | 5.0 |
| DT-SAM | 12 | 12.2 | 22.7 | 15.8 | 28.7 | 10.4 | 21.5 | 13.6 | 27.1 | 3.9 M | 7.6 G | 5.0 |
| PT-SAM | 12 | 30.2 | 63.4 | 41.3 | 76.5 | 32.1 | 66.4 | 41.1 | 74.3 | 0.13 M | 7.6 G | 5.0 |
| HQ-SAM | 12 | 31.9 | 65.5 | 42.9 | 77.1 | 33.6 | 68.4 | 42.2 | 75.9 | 5.1 M | 7.6 G | 4.8 |
| Stable-SAM | 1 | **44.8** | **76.4** | **50.5** | **81.1** | **43.3** | **75.6** | **48.6** | **79.4** | **0.08 M** | 7.6 G | 5.0 |

shown in Table 1) to 48.8/42.1 mIoU/mBIoU, accompanied by a low stability score of 39.5 ST. This is probably because SAM was trained with solely high-quality prompts, thus seriously suffers from the low-quality prompts during inference. Finetuning SAM's whole model greatly impairs performance, because it destroys the integrity of the highly-optimized SAM model and thus sacrificing the zero-shot segmentation generality. The other five SAM variants, namely HQ-SAM, DT-SAM, and PT-SAM, SAM with LoRA and SAM with Adapter, demonstrate relatively better stability in dealing with noisy boxes, which can be attributed to their long-term training on the HQSeg-44K dataset. Note our Stable-SAM can effectively address inaccurate box prompts, by enabling models to shift attention to target objects. Given a single-point prompt, both SAM and its variants exhibit the lowest accuracy and stability. This indicates they are adversely affected by the ambiguity problem arising from the use of a single-point prompt. Although, in most practical applications, users prefer minimal interaction with clear and consistent segmentation targets. Our method maintains much better performance and stability when handling ambiguous one-point prompt, owing to our deformable feature sampling and robust training strategy against ambiguity. When point prompts increase to 3, all methods perform much better, while other methods still under-perform compared with ours.

Segment Anything Model 2 (SAM 2) (Ravi et al., 2024) is a unified model for video and image-based promptable segmentation. SAM 2 outperforms SAM, due to its stronger backbone and larger pretraining dataset. However, SAM 2 still suffers significantly from low-quality prompts, owing to the overlooked segmentation stability problem. Our method can be seamlessly integrated into SAM 2 to enhance segmentation stability and performance under prompts of varying quality. Stable-SAM 2 exhibits substantial improvements in segmentation quality and stability, outperforming the original Stable-SAM model.

**Generalization Comparison on MS COCO and SGinW.** Table 3 presents the segmentation accuracy and stability when the models are generalized to MS COCO and SGinW with noisy boxes. Note that the DT-SAM performs the worst, probably due to overfitting on the training set, which compromises its ability to generalize to new datasets. Our method consistently surpasses all competitors, particularly in handling inaccurate boxes (N-Box 0.5–0.6), where all noisy boxes have an IoU range of 0.5–0.6 with the ground truth boxes. Note that our method has a minimal number of extra learnable parameters (0.08M) and can be quickly adapted to new datasets by just one training epoch. We also evaluate the learnable parameters, training memory and inference speed of our method. The results demonstrate that our approach is lightweight and efficient, with the negligible addition of 0.08 M parameters having no impact on the efficiency of the original SAM model.

**Comparison Based on Detector Predicted Box Prompts.** Existing zero-shot segmentation methods typically choose powerful object detection models to generate high-quality boxes as the input prompts, such as FocalNet-L-DINO (Zhang et al., 2023a; Yang et al., 2022). We also evaluate our method in such setting. Table 4 presents that our model achieves comparable performance as SAM and PT-SAM when using the FocalNet-L-DINO generated high-quality boxes as prompts. When using the R50-H-Deformable-DETR (Zhu et al., 2020) as the box prompt generator, our method achieves comparable performance as HQ-SAM. Note that training and implementing SOTA

Table 4: Comparison on MS COCO with the box prompts generated by SOTA detectors (FocalNet-L-DINO and R50-H-D-DETR) or noisy box prompts that overlap with the ground truth boxes, with IoU ranges of 0.5-0.6. All models (except for SAM) are trained on HQSeg-44K dataset.

| Model | FocalNet-L-DINO mAP | mAP$_{50}$ | R50-H-D-DETR mAP | mAP$_{50}$ | Noisy Box mAP | mAP$_{50}$ |
|---|---|---|---|---|---|---|
| SAM | 48.5 | 75.3 | 41.5 | 63.7 | 27.3 | 60.2 |
| PT-SAM | 48.6 | 75.5 | 41.7 | 64.2 | 30.2 | 63.4 |
| HQ-SAM | **49.5** | **75.7** | **42.4** | **64.5** | 31.9 | 65.5 |
| Ours | 48.3 | 74.8 | 42.2 | 64.0 | **44.8** | **76.4** |

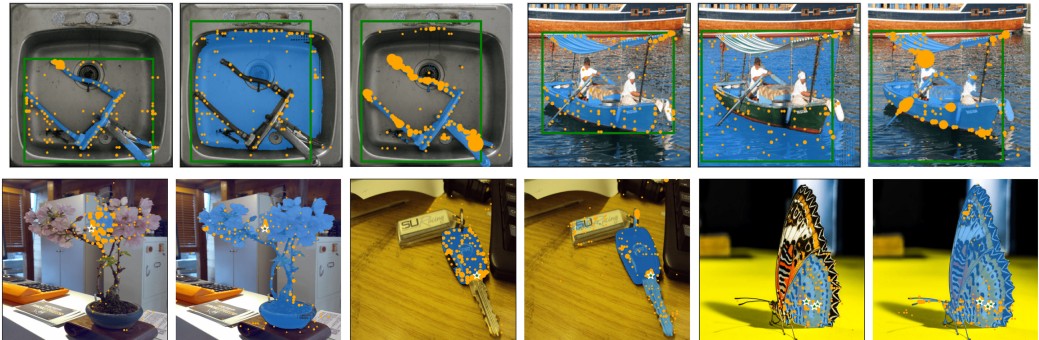

Figure 4: Visual results for box prompts (**1st** row), for point prompts (**2nd** row). Within each image group in the first three rows, the three figures represent the results of SAM with GT box prompt, SAM with noisy box prompt, and Stable-SAM with noisy box prompt, respectively. The last two rows display the results of SAM and Stable-SAM with point prompts.

detectors typically require large computational resources and the cross-domain generalization is still very challenging. In practice, users tend to leverage interactive tools to annotate objects for their personalized datasets. Our method substantially surpasses other competitors in such scenario, when the box can roughly indicate the target object.

## 5.2 ANALYSIS ON STABLE-SAM

We perform detailed analysis on Stable-SAM on its network modules, model scalability, low-shot generalization, point prompt quality, backbone variants, relation to other methods, and stability visualization. More experimental analysis are included in the supplementary material.

**Deformable Sampling Plugin.** Table 5 shows DSP can be trained with high-quality prompts (without RTS) to improve the performance and stability on low-quality prompts, although the model still exhibits some instability. When equipped with RTS, DSP can effectively learn to shift SAM's attention to target objects when subjecting to inaccurate prompts. To delve deeper into the deformable sampling mechanism, we visualize the sampled feature points and their corresponding attention weights. Figure 4 illustrates how our DSP effectively shifts model's attention to the target object, resulting in increased attention weights.

Table 5: Ablation study on deformable sampling plugin (DSP), dynamic routing plugin (DRP) and robust training strategy (RTS). All models (except for SAM) are trained on HQSeg-44K dataset.

| Model | Noisy Box | | | 1 Point | | |
|---|---|---|---|---|---|---|
| | mIoU | mBIoU | ST | mIoU | mBIoU | ST |
| SAM (baseline) | 48.8 | 42.1 | 39.5 | 43.3 | 37.4 | 45.1 |
| + DSP | 69.9 | 60.2 | 67.2 | 46.8 | 40.8 | 48.0 |
| + DSP + RTS | 81.7 | 73.5 | 81.6 | 75.9 | 67.5 | 70.6 |
| + DSP + DRP + RTS | **82.3** | **74.1** | **82.3** | **76.9** | **68.4** | **71.1** |

Consequently, the cross-attention module aggregates more target object features into the prompt tokens, thereby improving the segmentation quality of the target objects.

**Dynamic Routing Plugin.** We leverage DSP to dynamically route the model between the regular and deformable feature sampling modes, conditioned on the input prompt quality. We find that DRP tends to route more DSP features when dealing with worse prompts. The DSP routing weight $\alpha_1$ is increased from 0.469 to 0.614 when we change the point prompt from three points to one point. It indicates that lower-quality prompts rely more on DSP features to shift attention to the desirable regions. Table 5 shows that DRP can further improve model's performance, especially when handling the challenging one-point prompt scenario.

**Robust Training Strategy.** Robust training is critical for improving model's segmentation stability, but is usually overlooked in previous works. RTS can guide the model, including our DSP, to accurately segment target objects even when provided with misleading low-quality prompts. Table 6 shows that RTS substantially improves the segmentation stability of all the methods, albeit with a slight compromise in performance when dealing with high-quality prompts. Note that from the application of RTS, which can be attributed to our carefully designed deformable sampling plugin design.

**Model Scalability.** Our method solely calibrates SAM's mask attention by adjusting model's feature sampling locations and amplitudes using a minimal number of learnable parameters (0.08 M), while keeping the model architecture and parameters intact. This plugin design grants our method with excellent model scalability. Table 6 shows that our model can be rapidly optimized by just one training epoch, achieving comparable performance and stability. By scaling the training procedure to 12 epochs, our method achieves the best performance across all prompting settings. Additionally, our method can cooperate with other SAM variants. For instance, when combined with HQ-SAM, the performance and stability are further improved.

**Low-Shot Generalization.** Customized datasets with mask annotation are often limited, typically consisting of only hundreds of images. For a fair comparison, all methods in Table 7 are trained with RTS by 1 training epoch. Table 7 shows that HQ-SAM

Table 6: Ablation study on Robust Training Strategy (RTS) and model scalability. All models in this table (except for SAM) are trained on HQSeg-44K dataset by 12 training epochs, unless stated otherwise, with or without Robust Training Strategy (RTS). All models are evaluated on four HQ datasets with GT box prompt and noisy box prompt.

| Model | Groundtruth Box | | Noisy Box | | |
|---|---|---|---|---|---|
| | mIoU | mBIoU | mIoU | mBIoU | ST |
| SAM (baseline) | 79.5 | 71.1 | 48.8 | 42.1 | 39.5 |
| **Without RTS:** | | | | | |
| PT-SAM | 87.6 | 79.7 | 70.6 | 60.4 | 64.0 |
| HQ-SAM | **89.1** | 81.8 | 72.4 | 62.8 | 65.5 |
| Ours (1 epoch) | 87.4 | 80.0 | 69.6 | 60.0 | 66.5 |
| Ours (12 epochs) | **89.1** | **82.1** | **72.7** | **63.2** | **67.4** |
| **With RTS:** | | | | | |
| PT-SAM | 86.8 | 78.4 | 82.1 | 73.1 | 78.7 |
| HQ-SAM | 87.4 | 79.8 | 82.9 | 74.5 | 80.4 |
| Ours (1 epoch) | 86.0 | 78.4 | 82.3 | 74.1 | 82.3 |
| Ours (12 epochs) | 87.4 | 80.1 | 84.4 | 76.7 | 85.2 |
| HQ-SAM + Ours | **88.7** | **81.5** | **86.1** | **78.7** | **86.3** |

performs worst when trained with a limited number of images (220 or 440 images), which can be attributed to its potential overfitting problem caused by the relatively large learnable model parameters (5.1 M). In contrast, PT-SAM's better performance with minimal learnable parameters (0.13 M) further validates this hypothesis. Our plugin design, coupled with minimal learnable parameters, enables effective low-shot generalization, and thus achieves the best performance in such scenario.

**Discussion on Potential Segmentation Bias.** We emphasize that our method is designed without inherent bias towards large or small objects. If the prompt explicitly specifies the background, our method adapts accordingly, without constraining the model to prioritize the foreground. Although our method avoids introducing bias, we acknowledge that the model may still be influenced by dataset bias. For instance, if the training dataset predominantly contains foreground objects, the model may skew predictions towards the foreground, potentially neglecting background regions. Thus, we also highlight the flexibility of our method. If users wish to personalize segmentation targets, such as focusing on specific background regions, Stable-SAM can be easily fine-tuned to meet this requirement. This adaptability is a key strength, enabling our approach to effectively mitigate dataset bias and address a wide range of user needs and scenarios beyond typical foreground segmentation.

Table 7: Low-shot generalization comparison. All models are trained with RTS by 1 training epoch, with 220/440 train images. All models are evaluated on four HQ datasets with noisy box prompt and 1 point prompt.

| Model | Noisy Box | | | 1 Point | | |
|---|---|---|---|---|---|---|
| | mIoU | mBIoU | ST | mIoU | mBIoU | ST |
| SAM (baseline) | 48.8 | 42.1 | 39.5 | 43.3 | 37.4 | 45.1 |
| *220 train images:* | | | | | | |
| PT-SAM | 77.6 | 67.7 | 72.6 | 71.8 | 63.2 | 73.0 |
| HQ-SAM | 73.5 | 62.3 | 67.7 | 71.3 | 62.6 | 72.4 |
| Ours | **78.1** | **68.8** | **78.6** | **73.0** | **64.7** | **74.5** |
| *440 train images:* | | | | | | |
| PT-SAM | 78.6 | 69.0 | 74.4 | 73.1 | 65.5 | 71.1 |
| HQ-SAM | 77.4 | 67.1 | 75.6 | 71.7 | 61.8 | 68.8 |
| Ours | **79.5** | **70.3** | **81.7** | **75.6** | **67.5** | **71.5** |

## 6 CONCLUSION

In this paper, we present the first comprehensive analysis on SAM's segmentation stability across a wide range of prompt qualities. Our findings reveal that SAM's mask decoder tends to activate image features that are biased to the background or specific object parts. We propose the novel Stable-SAM to address this issue by calibrating solely SAM's mask attention, *i.e.*, adjusting the sampling locations and amplitudes of image feature using learnable deformable offsets, while keeping the original SAM model unchanged. The deformable sampling plugin (DSP) allows SAM to adaptively shift attention to the prompted target regions in a data-driven manner. The dynamic routing plugin (DRP) toggles SAM between deformable and regular grid sampling modes depending on the quality of the input prompts. Our robust training strategy (RTS) facilitates Stable-SAM to effectively adapt to prompts of varying qualities. Extensive experiments on multiple datasets validate the effectiveness and advantages of our Stable-SAM.

ACKNOWLEDGEMENTS

This research was supported in part by the Natural Science Foundation of Jiangsu Province (BK20241200), National Natural Science Foundation of China (62406140), Research Grant Council of the HKSAR (16201420), and Kuaishou Technology.

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
