# STABLE SEGMENT ANYTHING MODEL

**Qi Fan[1], Xin Tao[2,*], Lei Ke[3], Mingqiao Ye[4], Pengfei Wan[2], Di Zhang[2],**
**Yu-Wing Tai[5], Chi-Keung Tang[6]**
[1] Nanjing University, [2] Kuaishou Technology, [3] Carnegie Mellon University,
[4] EPFL, [5] Dartmouth College, [6] The Hong Kong University of Science and Technology

## 1 MORE IMPLEMENTATION DETAILS

**Training Details.** During training, we only train DSP and DRP on HQSeg-44K dataset while fixing the model parameters of the pre-trained SAM model. We train Stable-SAM on 8 NVIDIA Tesla V100 GPUs with a total batch size of 32, using Adam optimizer with zero weight decay and 0.001 learning rate. The training images are augmented using large-scale jittering (Ghiasi et al., 2021). The input prompts are randomly sampled from mixed prompt types, including ground truth bounding boxes, randomly sampled points (1, 3, 5, 10 positive points randomly selected from the ground truth mask), noisy boxes (generated by adding noise (noise scale 0.4) to the ground truth bounding boxes, where we ensure the generated noisy boxes have at least 0.5 overlap IoU with the ground truth boxes), and coarse masks (generated by adding Gaussian noise in the boundary regions of the ground truth masks). The model is optimized using cross entropy loss and dice loss (Milletari et al., 2016).

**Inference Details.** We follow the same inference pipeline of the original SAM. The mask decoder first predicts a small mask in $256 \times 256$ spatial resolution for each prompt, which is then up-sampled to the original resolution $1024 \times 1024$ as the output mask.

**Evaluation Metrics.** We select suitable evaluation metrics depending on testing datasets, *i.e.*, 1) mask mIoU, boundary mBIoU and ST for DIS, ThinObject-5K, COIFT, and HR-SOD, which usually contain only one object in each image; 2) mask mAP and $mAP_{50}$ for COCO and SGinW, which usually contain multiple objects in each image.

**Occlusion Image Synthesis.** For each training image, we randomly select another image with random scale jittering and random horizontal flipping augmentation. Then we select a random object from the selected image as the "occluder" and paste it onto the training image to occlude the "occludee" object. Specifically, we ensure the center of the occluder is strategically placed within the bounding box of the occluded object in the training image. Finally, we remove the fully occluded objects and update the ground-truth mask annotations of the partially occluded objects. The occlusion image synthesis is enabled with a probability of 0.5 in the training stage.

**More implementation details of LoRA and adapters in SAM.** We introduce Adapter/LoRA modules to the feed-forward network (FFN) of each ViT layer in SAM's encoder for tuning. During training, we fine-tune only the adapter/LoRA modules and SAM's prediction layer, with all other parameters frozen. In line with AdaptFormer (Chen et al., 2022), the adapters consist of small bottleneck layers inserted in parallel into the FFN, containing two MLPs and a GELU activation function between them. The bottleneck's middle dimension is set to 64 to balance model performance and computational efficiency. In line with LoRA (Hu et al., 2022), the module uses an encoder-decoder structure to impose a low-rank constraint on FFN weight updates, injecting small trainable rank decomposition matrices into each layer. In our experiments, the rank of LoRA is set to 4 for efficiency and performance optimization. All other experimental settings remain the same as those of the baseline and full model.

**More implementation details for SAM-based interactive segmentation.** In the SAM-based interactive segmentation experiments, setting the hyperparameter 'multimask_output = False' will yield better performance, especially on SBD dataset. For a fair comparison, we use the default 'multimask_output = True' setting to align with the SAM-based interaction segmentation implementations potentially adopted by other works.

---

*Corresponding Author: Xin Tao

Table 1: Comparison on Multi-domain Evaluation of Semantic Segmentation (MESS) benchmark, consisting of 22 downstream datasets, 448 classes, and 25,079 images.

| Model | Prompt | General | Earth Monitoring | Medical Sciences | Engineering | Agriculture & Biology | Mean |
|---|---|---|---|---|---|---|---|
| SAM | | 46.51 | 42.31 | 55.92 | 51.57 | 57.43 | 49.67 |
| HQ-SAM | Oracle Point | 44.05 | 40.82 | **61.0** | **54.18** | 53.92 | 49.58 |
| Stable-SAM (Ours) | | **49.10** | **43.23** | 54.45 | 53.52 | **60.92** | **51.15** |
| SAM | | 77.85 | **73.03** | 64.89 | 73.03 | **86.75** | **74.74** |
| HQ-SAM | Oracle Box | 76.63 | 68.03 | **65.1** | **74.78** | 85.34 | 73.43 |
| Stable-SAM (Ours) | | **77.91** | 72.84 | 62.93 | 73.12 | 85.63 | 74.21 |
| SAM | | 40.21 | 36.74 | 53.20 | 47.15 | 49.75 | 44.34 |
| HQ-SAM | Random Point | 38.81 | 36.62 | **58.27** | **51.43** | 44.51 | 44.92 |
| Stable-SAM (Ours) | | **42.43** | **37.30** | 50.89 | 48.96 | **53.42** | **45.49** |
| SAM | | 55.97 | 55.12 | 62.45 | 62.34 | 64.26 | 59.24 |
| HQ-SAM | Noisy Box | 52.02 | 47.15 | **62.57** | 62.02 | 60.56 | 55.81 |
| Stable-SAM (Ours) | | **59.82** | **58.86** | 62.19 | **64.56** | **68.84** | **62.13** |

Table 2: Dataset and comparison details on MESS benchmark. Models are prompted with noisy boxes. "HQ" denotes HQ-SAM.

| Dataset | Domain | Sensor type | Mask size | # Classes | # Images | Task | SAM | HQ | Ours |
|---|---|---|---|---|---|---|---|---|---|
| BDD100K (Yu et al., 2020) | | Visible spectrum | Medium | 19 (Medium) | 1,000 | Driving | 48.9 | 43.66 | **51.84** |
| Dark Zurich (Sakaridis et al., 2019) | | Visible spectrum | Medium | 20 (Medium) | 50 | Driving | 54.34 | 50.85 | **56.81** |
| MHP v1 (Li et al., 2017) | General | Visible spectrum | Small | 19 (Medium) | 980 | Body parts | 66.64 | 62.63 | **69.11** |
| FoodSeg103 (Ghosh et al., 2021) | | Visible spectrum | Medium | 104 (Many) | 2,135 | Ingredients | 57.98 | 56.22 | **64.12** |
| ATLANTIS (Erfani et al., 2022) | | Visible spectrum | Small | 56 (Many) | 1,295 | Maritime | 56.23 | 48.7 | **60.04** |
| DRAM (Cohen et al., 2022) | | Visible spectrum | Medium | 12 (Medium) | 718 | Paintings | 51.73 | 50.06 | **57.03** |
| iSAID (Waqas Zamir et al., 2019) | | Visible spectrum | Small | 16 (Medium) | 4,055 | Objects | 65.2 | 62.6 | **67.46** |
| ISPRS Potsdam (Khoshelham et al., 2017) | Earth Monitoring | Multispectral | Small | 6 (Few) | 504 | Land use | 47.32 | 38.43 | **50.92** |
| WorldFloods (Mateo-Garcia et al., 2021) | | Multispectral | Medium | 3 (Binary) | 160 | Floods | 57.61 | 49.46 | **60.81** |
| FloodNet (Rahnemoonfar et al., 2021) | | Visible spectrum | Medium | 10 (Few) | 5,571 | Floods | 51.85 | 39.82 | **58.31** |
| UAVid (Lyu et al., 2020) | | Visible spectrum | Small | 8 (Few) | 840 | Objects | 53.62 | 45.43 | **56.82** |
| Kvasir-Inst. (Jha et al., 2021) | | Visible spectrum | Medium | 2 (Binary) | 118 | Endoscopy | 83.62 | 81.32 | **85.87** |
| CHASE DB1 (Fraz et al., 2012) | Medical Sciences | Microscopic | Small | 2 (Binary) | 20 | Retina scan | 34.59 | **37.5** | 33.88 |
| CryoNuSeg (Mahbod et al., 2021) | | Microscopic | Small | 2 (Binary) | 30 | WSI | 72.87 | 72.78 | **73.09** |
| PAXRay-4 (Seibold et al., 2022) | | Electromagnetic | Large | 4x2 (Binary) | 180 | X-Ray | **58.72** | 58.66 | 55.94 |
| Corrosion CS (Bianchi & Hebdon, 2021) | | Visible spectrum | Medium | 4 (Few) | 44 | Corrosion | 53.05 | 51.86 | **55.47** |
| DeepCrack (Liu et al., 2019) | Engineering | Visible spectrum | Small | 2 (Binary) | 237 | Cracks | 56.36 | **60.84** | 58.2 |
| ZeroWaste-f (Bashkirova et al., 2022) | | Visible spectrum | Medium | 5 (Few) | 929 | Conveyor | 70.23 | 69.52 | **74.01** |
| PST900 (Shivakumar et al., 2020) | | Electromagnetic | Small | 5 (Few) | 288 | Thermal | 69.71 | 65.88 | **70.55** |
| SUIM (Islam et al., 2020) | | Visible spectrum | Medium | 8 (Few) | 110 | Underwater | 56.98 | 47.51 | **62.35** |
| CUB-200 (Welinder et al., 2010) | Agriculture & Biology | Visible spectrum | Medium | 201 (Many) | 5,794 | Bird species | 56.3 | 54.54 | **64.57** |
| CWFID (Haug & Ostermann, 2015) | | Visible spectrum | Small | 3 (Few) | 21 | Crops | 79.49 | **79.63** | 79.61 |
| Mean IoU | – | – | – | – | – | – | 59.24 | 55.81 | **62.13** |

## 2 MORE EXPERIMENTAL RESULTS

### 2.1 MULTI-DOMAIN EVALUATION OF SEMANTIC SEGMENTATION (MESS)

The recently released Multi-domain Evaluation of Semantic Segmentation (MESS) (Blumenstiel et al., 2023) is a large-scale benchmark for holistic analysis of zero-shot segmentation performance. MESS consists of 22 downstream tasks, a total of 448 classes, and 25079 images, covering a wide range of domain-specific datasets in the fields of earth monitoring, medical sciences, engineering, agriculture and biology and other general domains. We evaluate SAM (Kirillov et al., 2023), HQ-SAM (Ke et al., 2023) and our Stable-SAM on MESS benchmark using the official MESS evaluation code, and report the mean of class-wise intersection over union (mIoU).

Following MESS's model settings, our Stable-SAM selects the first mask of the predicted multiple masks as the output. For a fair comparison, our Stable-SAM follows HQ-SAM to fuse the SAM's original prediction map into our predicted segmentation map. We provide four prompt types for evaluation. The *oracle point* refers to a single point sampled from the ground-truth mask using the point sampling approach RITM (Sofiiuk et al., 2022). The *random point* refers to a single point

Table 3: User study of noisy boxes for realistic application scenarios.

| Model | Noisy Box | | | 1 Point | | |
|---|---|---|---|---|---|---|
| | mIoU | mBIoU | ST | mIoU | mBIoU | ST |
| SAM (baseline) | 48.8 | 42.1 | 39.5 | 58.3 | 51.7 | 49.3 |
| HQ-SAM | 72.4 | 62.8 | 65.5 | 76.1 | 66.4 | 69.2 |
| Ours | 82.3 | 74.1 | 82.3 | 84.2 | 76.3 | 84.0 |

Table 4: Comparison on MS COCO and four HQ datasets for different backbone variants.

| Model | Epoch | MS COCO | | Four HQ Datasets | | | | | | Params (M) | | FPS | Mem. |
|---|---|---|---|---|---|---|---|---|---|---|---|---|---|
| | | N-Box (0.5-0.6) | | Noisy Box | | | 1 Point | | | | | | |
| | | mAP | mAP$_{50}$ | mIoU | mBIoU | mSF | mIoU | mBIoU | mSF | Total | Trainable | | |
| SAM-Huge | - | 25.6 | 56.8 | 50.1 | 43.2 | 40.4 | 44.5 | 38.3 | 46.5 | 2446 | 2446 | 3.5 | 10.3G |
| HQ-SAM-Huge | 12 | 30.0 | 62.6 | 75.2 | 65.5 | 69.3 | 48.0 | 41.1 | 49.5 | 2452.1 | 6.1 | 3.4 | 10.3G |
| Stable-SAM-Huge | 1 | **43.9** | **75.7** | **81.8** | **73.5** | **82.3** | **77.2** | **68.9** | **74.6** | 2446.08 | 0.08 | 3.5 | 10.3G |
| SAM-Large | - | 27.3 | 60.2 | 48.8 | 42.1 | 39.5 | 43.3 | 37.4 | 45.1 | 1191 | 1191 | 5.0 | 7.6G |
| HQ-SAM-Large | 12 | 31.9 | 65.5 | 72.4 | 62.8 | 65.5 | 43.2 | 44.6 | 37.4 | 1196.1 | 5.1 | 4.8 | 7.6G |
| Stable-SAM-Large | 1 | **44.8** | **76.4** | **82.3** | **74.1** | **82.3** | **76.9** | **68.4** | **71.1** | 1191.08 | 0.08 | 5.0 | 7.6G |
| SAM-Base | - | 19.7 | 49.2 | 41.6 | 35.8 | 33.4 | 35.1 | 29.2 | 36.7 | 358 | 358 | 10.1 | 5.1G |
| HQ-SAM-Base | 12 | 24.7 | 56.1 | 68.7 | 59.1 | 63.2 | 40.6 | 35.7 | 42.7 | 362.1 | 4.1 | 9.8 | 5.1G |
| Stable-SAM-Base | 1 | **31.2** | **63.3** | **74.7** | **64.8** | **75.9** | **68.9** | **59.5** | **67.1** | 358.08 | 0.08 | 10.1 | 5.1G |

randomly sampled from the ground-truth mask of the target object. The *oracle box* refers to a single box tightly enclosing the ground-truth mask of the target object. The *noisy box* refers to a single box generated by adding noise (noise scale 0.4) to the *oracle box*.

Table 1 tabulates the zero-shot semantic segmentation performance comparison on MESS. Our Stable-SAM performs best when prompted with *oracle point*, *random point* and *noisy box*, and achieves comparable performance when provided with *oracle box*. Our competitive performance on the large-scale MESS benchmark further consolidates the powerful zero-shot generalization ability inherent in our Stable-SAM. Table 2 shows the dataset and comparison details on 22 tasks of MESS benchmark. Our Stable-SAM performs best on 19 out of 22 datasets.

## 2.2 COMPARISON BASED ON USER-ANNOTATED BOX PROMPTS

We conduct a user study to provide a more realistic evaluation of our method. Five participants are asked to provide box annotations for the highlighted target object in each image. Participants are instructed to complete each box annotation within 5 seconds to ensure consistency throughout the process. Annotations are collected using the Label Studio platform. The user-annotated boxes are subsequently used as prompts to evaluate the performance of each segmentation method. Table 3 shows that user-annotated boxes provide better segmentation performance across all methods compared to generated noisy boxes. We also assess the quality of the user-annotated boxes by comparing them to the ground truth boxes derived from the mask annotations. The average IoU between user-annotated boxes and ground truth boxes is approximately 0.753, indicating that user-annotated boxes are more accurate than generated noisy boxes. Under the user-provided box prompt, our method continues to outperform other methods by a large margin.

## 2.3 BACKBONE VARIANTS

Table 4 tabulates the performance comparison on different backbone variants. Our Stable-SAM consistently performs better than other methods on all backbone variants.

## 2.4 ABLATION STUDIES ON THE NUMBER OF PROMPT POINTS

Table 5 shows the performance curve of SAM, PT-SAM and Stable-SAM when handling various number of prompt points. We also show the performance standard deviation to indicate the segmenta-

Table 5: Ablation studies on the number of prompt points. All models (except for SAM) are trained on HQSeg-44K dataset, and evaluated on four HQ datasets with various prompt points.

| # of Points | 1 | 3 | 5 | 7 | 10 | 15 | 20 |
|---|---|---|---|---|---|---|---|
| SAM | $43.3_{\pm1.8}$ | $78.7_{\pm0.8}$ | $83.3_{\pm0.7}$ | $84.1_{\pm0.7}$ | $84.8_{\pm0.6}$ | $85.3_{\pm0.6}$ | $85.7_{\pm0.5}$ |
| PT-SAM | $43.0_{\pm1.9}$ | $80.1_{\pm0.7}$ | $84.3_{\pm0.6}$ | $85.2_{\pm0.6}$ | $85.6_{\pm0.4}$ | $86.1_{\pm0.4}$ | $86.3_{\pm0.3}$ |
| Stable-SAM | $76.9_{\pm0.8}$ | $84.0_{\pm0.6}$ | $85.8_{\pm0.5}$ | $86.5_{\pm0.4}$ | $87.0_{\pm0.4}$ | $87.5_{\pm0.3}$ | $87.8_{\pm0.3}$ |

Table 6: The DRP routing weight $\alpha_1$ is affected by the number of the prompt points. Fewer prompt points needs more deformable attention indicated by larger $\alpha_1$. The models is trained on HQSeg-44K dataset and evaluated on four HQ datasets.

| # of Points | 1 | 2 | 3 | 4 | 5 | 10 | 20 |
|---|---|---|---|---|---|---|---|
| $\alpha_1$ | 0.614 | 0.552 | 0.469 | 0.427 | 0.398 | 0.371 | 0.359 |

Table 7: The DRP routing weight $\alpha_1$ is affected by the number of the prompt points. Fewer prompt points needs more deformable attention indicated by larger $\alpha_1$. The models is trained on HQSeg-44K dataset and evaluated on four HQ datasets.

| $\alpha_1$ | 0.0 | 0.2 | 0.4 | 0.5 | 0.6 | 0.8 | 1.0 |
|---|---|---|---|---|---|---|---|
| 1 point | 43.3 | 73.2 | 75.9 | 76.5 | 76.9 | 76.5 | 76.0 |
| 10 points | 84.8 | 86.1 | 87.0 | 86.9 | 86.7 | 86.7 | 86.5 |

tion stability. The results show that our Stable-SAM has larger performance gains when handling lower-quality prompts, *i.e.*, fewer prompt points.

## 2.5 ANALYSIS ON THE DRP ROUTING WEIGHT $\alpha_1$

Table 6 shows the DRP routing weight $\alpha_1$ is increased from 0.469 to 0.614 when we change the point prompt from three points to one point. It indicates that lower-quality prompts rely more on DSP features to shift attention to the desirable regions.

We further conduct additional experiments to manipulate the output strength of the DRP and examine its impact on segmentation quality for different prompt qualities. Specifically, we evaluate segmentation performance at different $\alpha_1$ values for both ambiguous (one point) and precise (ten points) prompts. For a fair comparison, we use the same set of point prompts for each $\alpha_1$ value.

Table 7 show that larger $\alpha_1$ values (indicating stronger DSP activation) lead to better segmentation performance for the ambiguous one point prompt. This suggests that increasing the output strength of the DRP is particularly beneficial for less informative prompts. In contrast, segmentation performance is less sensitive to variations in $\alpha_1$ for precise prompts, suggesting that when prompts provide clearer guidance, the system achieves satisfactory results even with lower activation levels.

## 3 RELATION TO OTHER METHODS

**Deformable Attention.** Our method is unique in its idea and design on solely adjusting the feature sampling locations and amplitudes by training the offset network, without involving the original model parameters. In contrast, conventional deformable attention methods (Dai et al., 2017; Xia et al., 2022) train both the offset network and original network parameters, which is undesirable when adapting powerful foundation models in deployment, especially in finetuning large foundation models. Figure 1 shows the difference between our deformable sampling plugin and conventional deformable attention.

We apply the conventional deformable attention in our Stable-SAM by finetuning the mask decoder during training. Table 8 shows that the conventional deformable attention (Stable-SAM (finetuning

Table 8: Comparison on MS COCO and four HQ datasets for different Stable-SAM variants. The "finetuning decoder" denotes finetuning the mask decoder when training Stable-SAM.

| | MS COCO | | Four HQ Datasets | | | | | |
| | N-Box (0.5-0.6) | | Noisy Box | | | 1 Point | | |
| Model | mAP | mAP$_{50}$ | mIoU | mBIoU | mSF | mIoU | mBIoU | mSF |
|---|---|---|---|---|---|---|---|---|
| SAM | 27.3 | 60.2 | 48.8 | 42.1 | 39.5 | 43.3 | 37.4 | 45.1 |
| Stable-SAM (finetuning decoder) | 25.7 | 56.5 | 78.5 | 69.2 | 79.9 | 76.0 | 67.1 | **78.2** |
| Stable-SAM (spatial attention) | 29.8 | 64.9 | 69.3 | 59.8 | 57.8 | 51.6 | 44.5 | 49.1 |
| Stable-SAM | **44.8** | **76.4** | **82.3** | **74.1** | **82.3** | **76.9** | **68.4** | 71.1 |

Figure 1: Method difference between our deformable sampling plugin and conventional deformable attention.

decoder)) exhibits the worst generalization ability on MS COCO, even worse than the original SAM model. This further validates the necessity and better performance of our deformable sampling plugin paradigm, *i.e.*, adapting the foundation model by only adjusting the feature sampling locations and amplitudes, while fixing the original model features and parameters.

**Spatial Attention.** The spatial attention (Woo et al., 2018; Hou et al., 2021) can adjust the image spatial feature weights, and thus can be regarded as a soft feature sampling method. We directly replace DSP with spatial attention in our Stable-SAM to investigate if spatial attention offers comparable effectiveness. Table 8 shows that spatial attention performs much worse than our DSP, although it consistently improves the segmentation performance and stability on all datasets. This indicates that simply adjusting the feature weights is insufficient to adapt SAM for handling suboptimal prompts.

## 4 MORE DISCUSSIONS

### 4.1 DISCUSSION ON MODEL SCALABILITY TO NOISY TRAINING DATA

Increasing the amount of training data could improve the model's robustness and generalization. More noisy data could help the model learn to handle a wider variety of input prompts, especially in real-world scenarios where user-provided prompts are often imprecise or ambiguous. As the model is exposed to more diverse and challenging inputs, it may become better at distinguishing relevant features and handling uncertainty in segmentation. However, introducing excessive noisy data could cause the model to overfit to the noise, leading to instability in some cases. Our method, which dynamically calibrates attention based on prompt quality, mitigates some risks of noisy data by guiding the model to focus on relevant image regions without being overwhelmed by noise.

We also understand the concerns about the scalability of our method when scaling up the training data. While our method is designed to efficiently handle noisy data with limited resources, it can still benefit from larger training sets. As the training dataset grows, we can fine-tune additional parameters of Stable-SAM to further improve segmentation stability. Our method's lightweight design, with only 0.08M learnable parameters, allows for fine-tuning additional parameters without significantly increasing model complexity or computational cost. This additional fine-tuning of more parameters applied to larger datasets, can further boost performance and stability, as the model adapts more precisely to the increased diversity of data and ambiguous prompts. This highlights the flexibility of our approach, which performs well with limited data but can also benefit from additional fine-tuning of more parameters when the training dataset increases in size and noise.

## 4.2 DISCUSSION ON THE MOTIVATION BEHIND DSP (DYNAMIC SAMPLING PLUGIN)

**Problem motivation.** The primary motivation for adjusting the attention sampling positions through learnable offsets is to overcome the inherent limitations of traditional fixed-grid attention, especially in the presence of noisy prompts. In the original SAM architecture, the mask decoder uses a regular grid sampling strategy, assuming that the initial prompt provides an accurate indication of the target region. However, in real-world scenarios where prompts are imprecise, fixed-grid sampling can cause the attention mechanism to focus on unintended areas, often capturing background or partially correct features.

**Method motivation.** Our approach introduces learnable offsets to adapt the sampling positions dynamically, enabling attention to more precisely target the correct regions, even with ambiguous or noisy prompts. The learnable offsets enable the attention mechanism to "deform" the regions it focuses on in a data-driven manner, akin to how a person might adjust their focus when given unclear instructions. Essentially, the model learns from diverse training examples how to "interpret" and refine ambiguous or noisy prompts, using the learned offsets to direct attention to the most contextually relevant regions of the image.

### 4.3 DISCUSSION ON THE CHOICE OF PARAMETER-EFFICIENT FINE-TUNING (PEFT) AND FULL RETRAINING

The decision to use PEFT rather than retraining all components of SAM primarily depends on the scale and type of the training dataset. SAM was pretrained on the SA-1B dataset, containing 1 billion masks and 11 million images, which endows it with a generalized ability to segment a wide range of objects across various domains. In contrast, our downstream dataset, HQSeg-44K, contains only 44,000 images. Due to the significant difference in dataset scale, PEFT is a strategic choice to prevent overfitting while effectively adapting the pre-trained SAM to our target segmentation tasks with a small training budget.

Numerous studies have demonstrated that PEFT methods are particularly effective when adapting large pretrained models to downstream tasks with limited data. This approach retains the generalizable features learned during pre-training, fine-tuning only a small subset of additional parameters. This paradigm is well-established in parameter-efficient fine-tuning research (as summarized in the PEFT surveys (Han et al., 2024; Xin et al., 2024)). By using PEFT, we aim to retain SAM's foundational representation power while mitigating overfitting, which is more likely to occur if we were to retrain the entire model on a relatively small dataset.

In our case, finetuning only the DSP and DRP modules ensures that SAM retains its generality and adaptability while becoming more robust to noisy and ambiguous prompts, maintaining computational efficiency. In contrast, retraining SAM's entire model compromises its integrity and significantly impairs performance, as shown by the large performance drop of FT-SAM (finetuning SAM's whole model) in Table 2 of the main paper.

## 5 STABILITY VISUALIZATION

Figure 2-12 show extensive visualization comparisons between SAM and Stable-SAM, under box, 1-point and 3-points prompts of diverse qualities. We also visualize the image activation map for the token-to-image cross-attention in SAM's second mask decoder layer to better understand its response to low-quality prompts. The important features are highlighted by the orange circles, with larger radius indicating higher attention score. SAM yields unsatisfactory segmentation results when provided with low-quality prompts, and even a minor prompt modification leads to unstable segmentation output. In contrast, our Stable-SAM produces consistent and accurate mask predictions even under prompts of diverse qualities, by shifting more feature sampling attention to the target object.

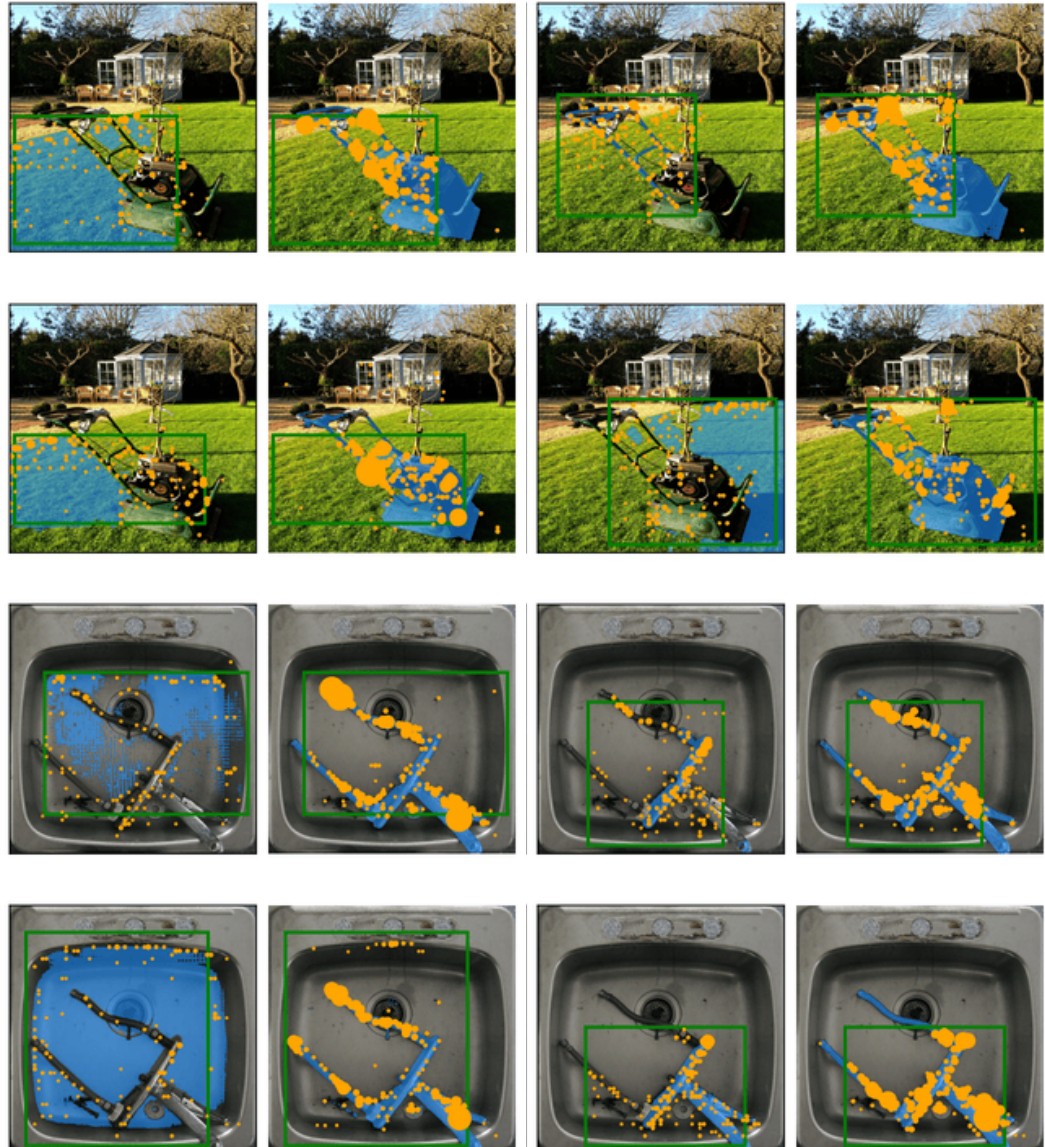

Figure 2: Visual results for box prompts. Within each image pair given the same prompt (green box), the subfigures represent the results of SAM and Stable-SAM, respectively.

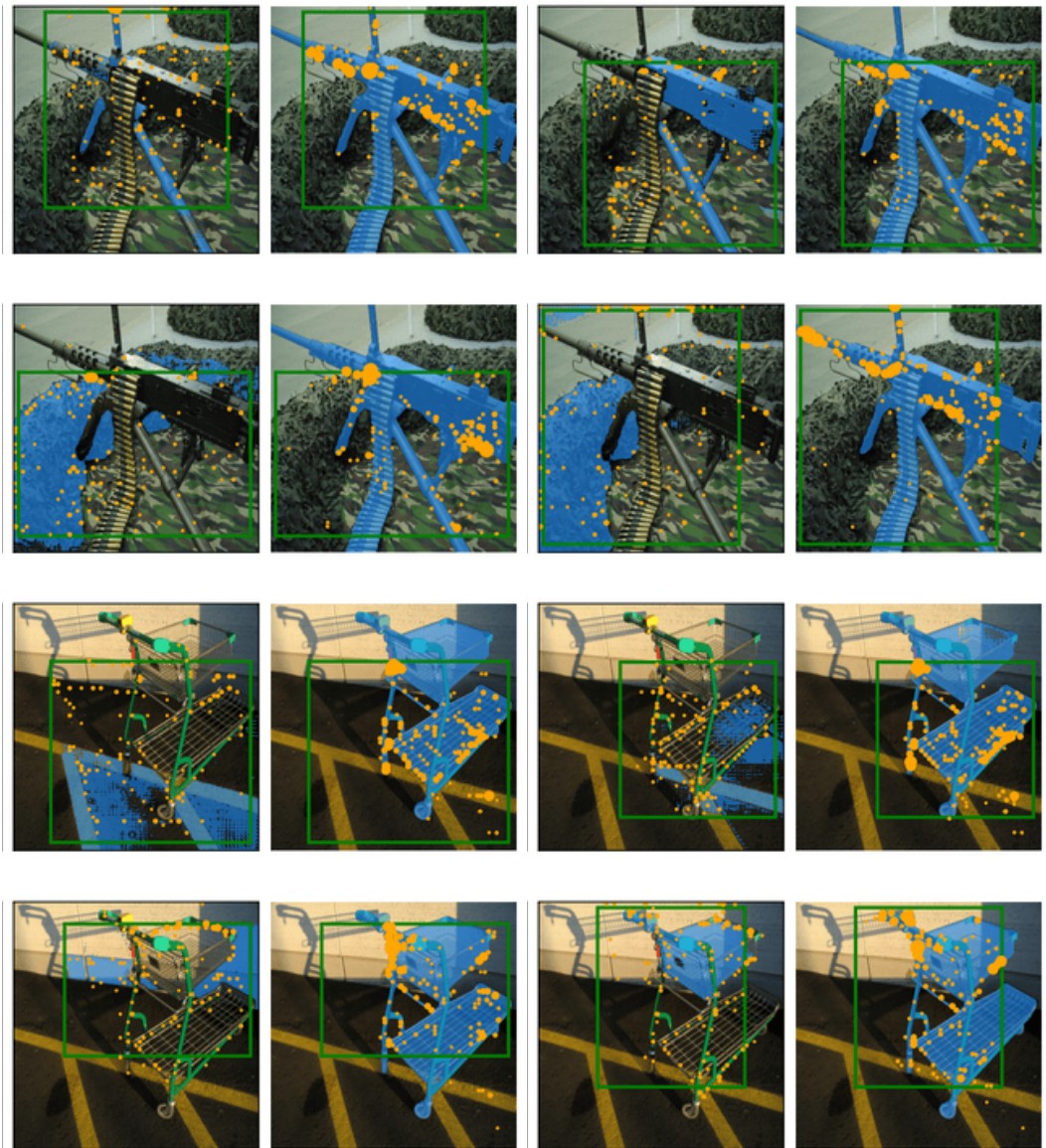

Figure 3: Visual results for box prompts.

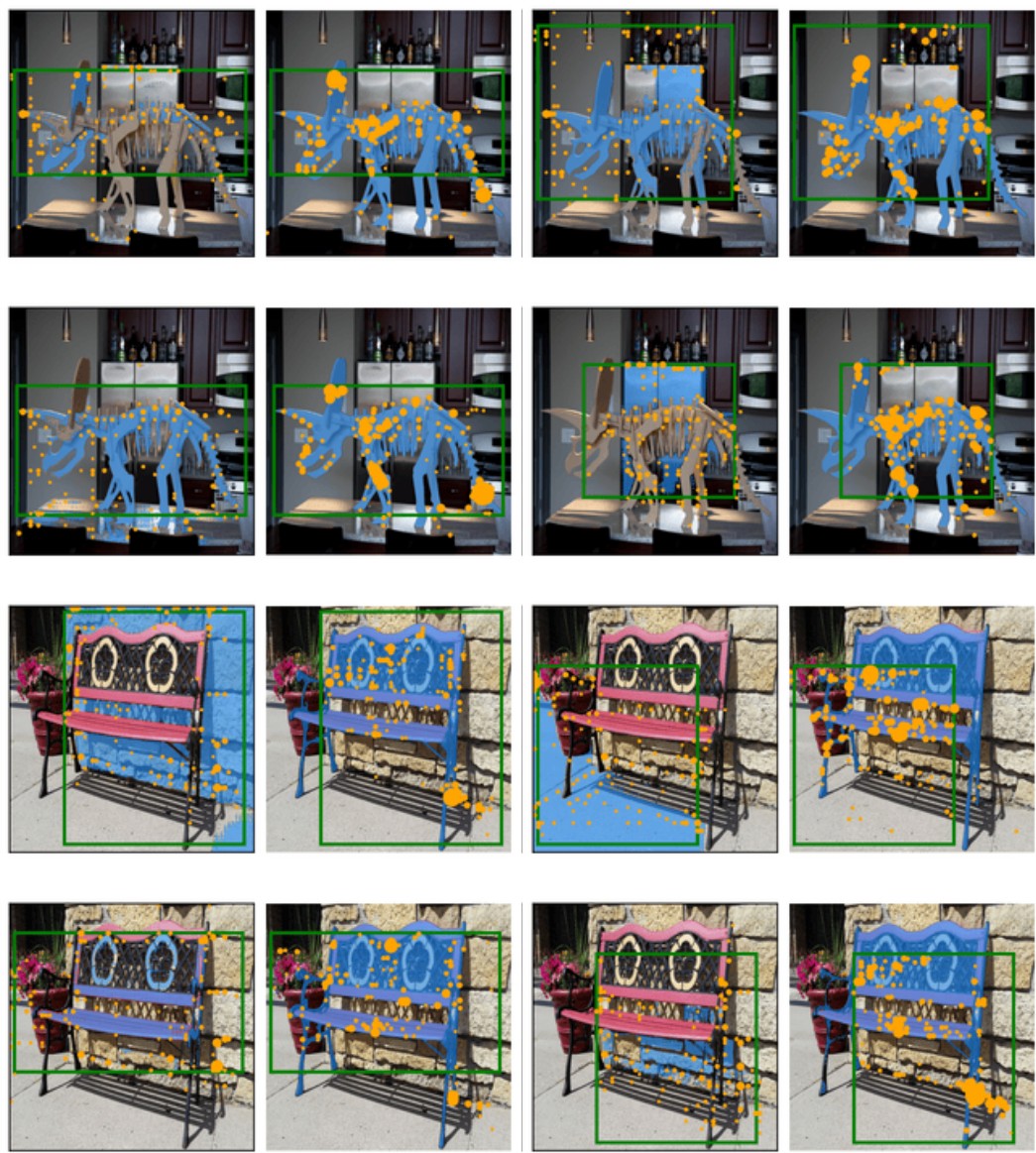

Figure 4: Visual results for box prompts.

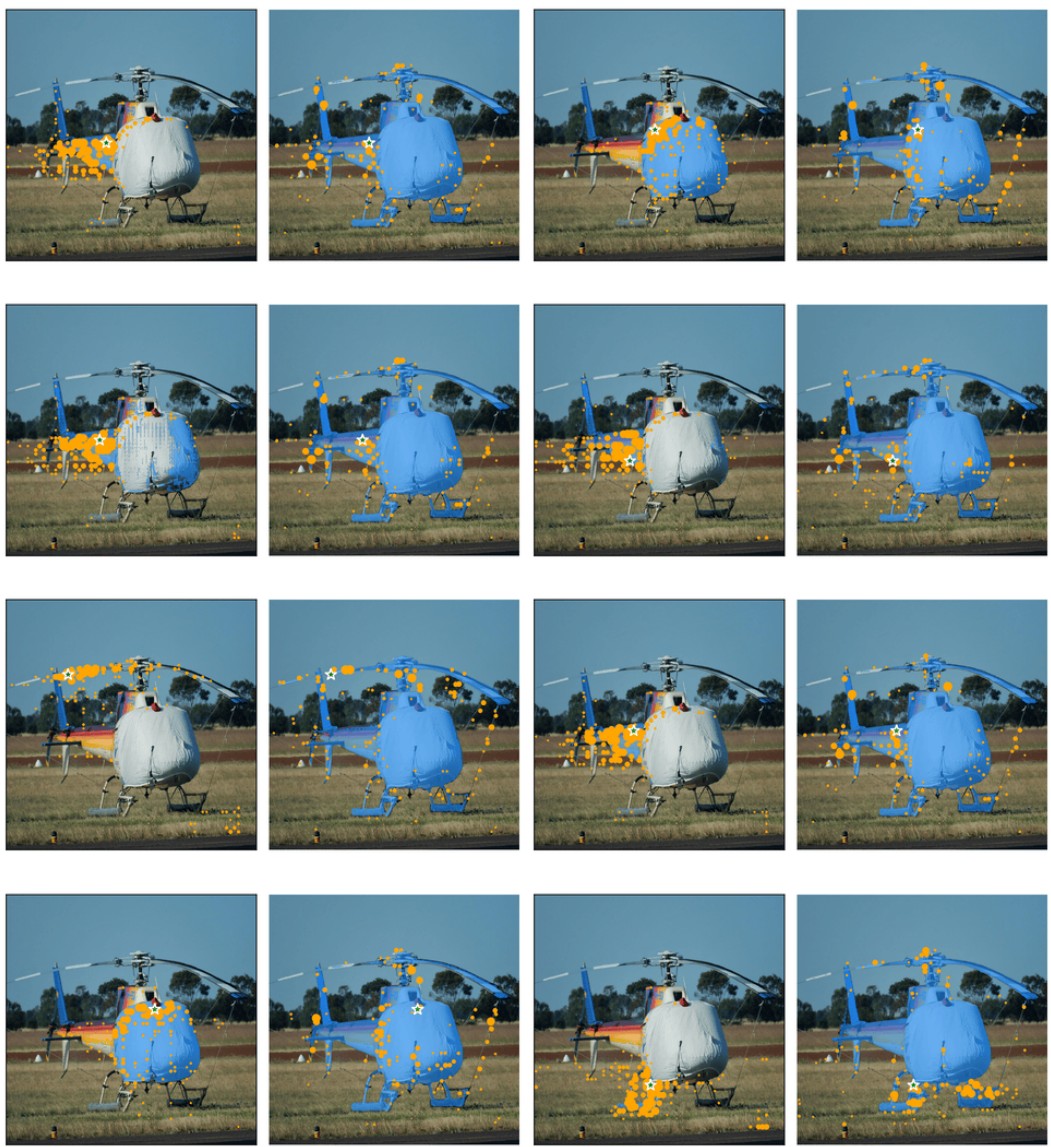

Figure 5: Visual results for 1-point prompt.

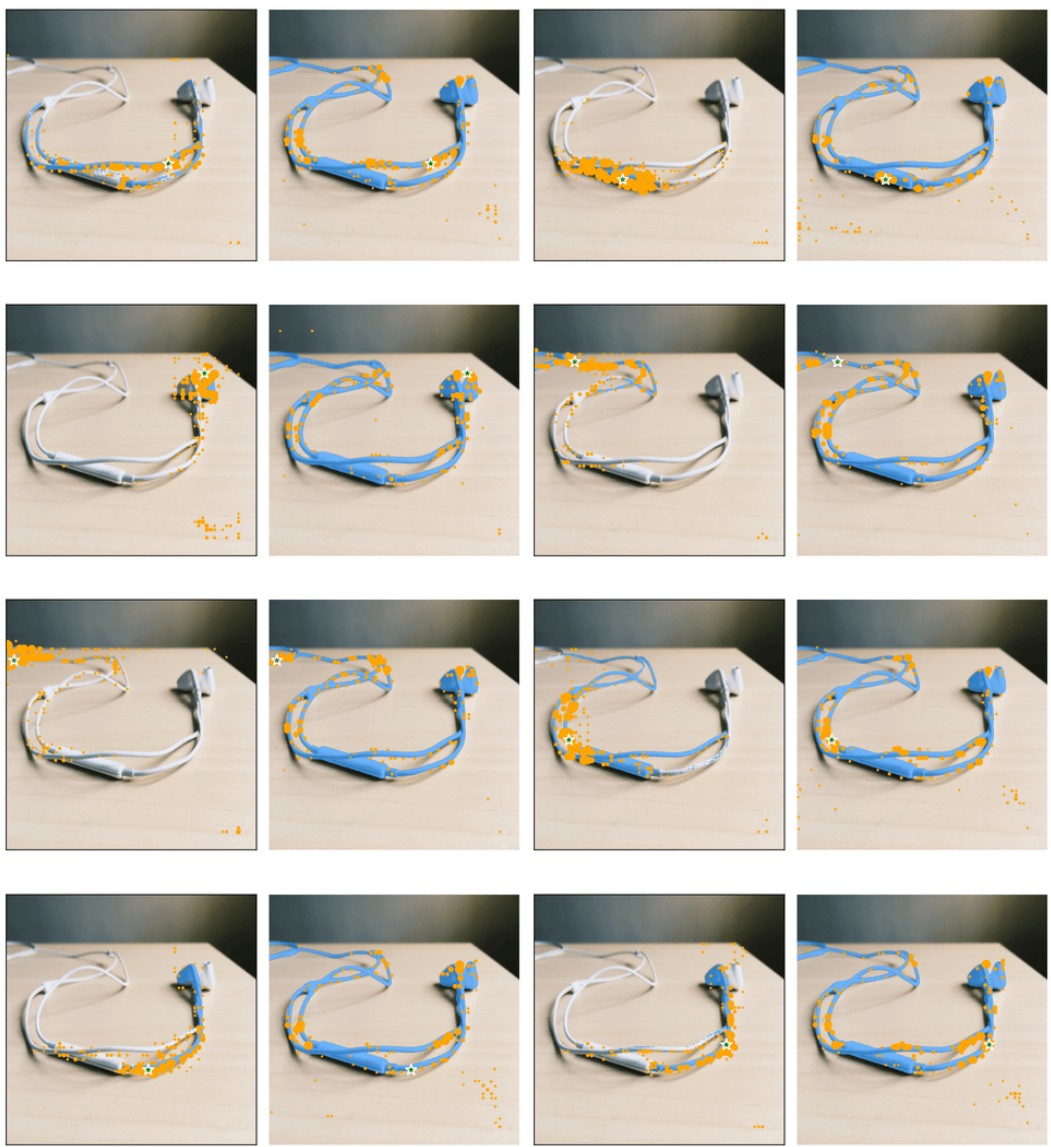

Figure 6: Visual results for 1-point prompt.

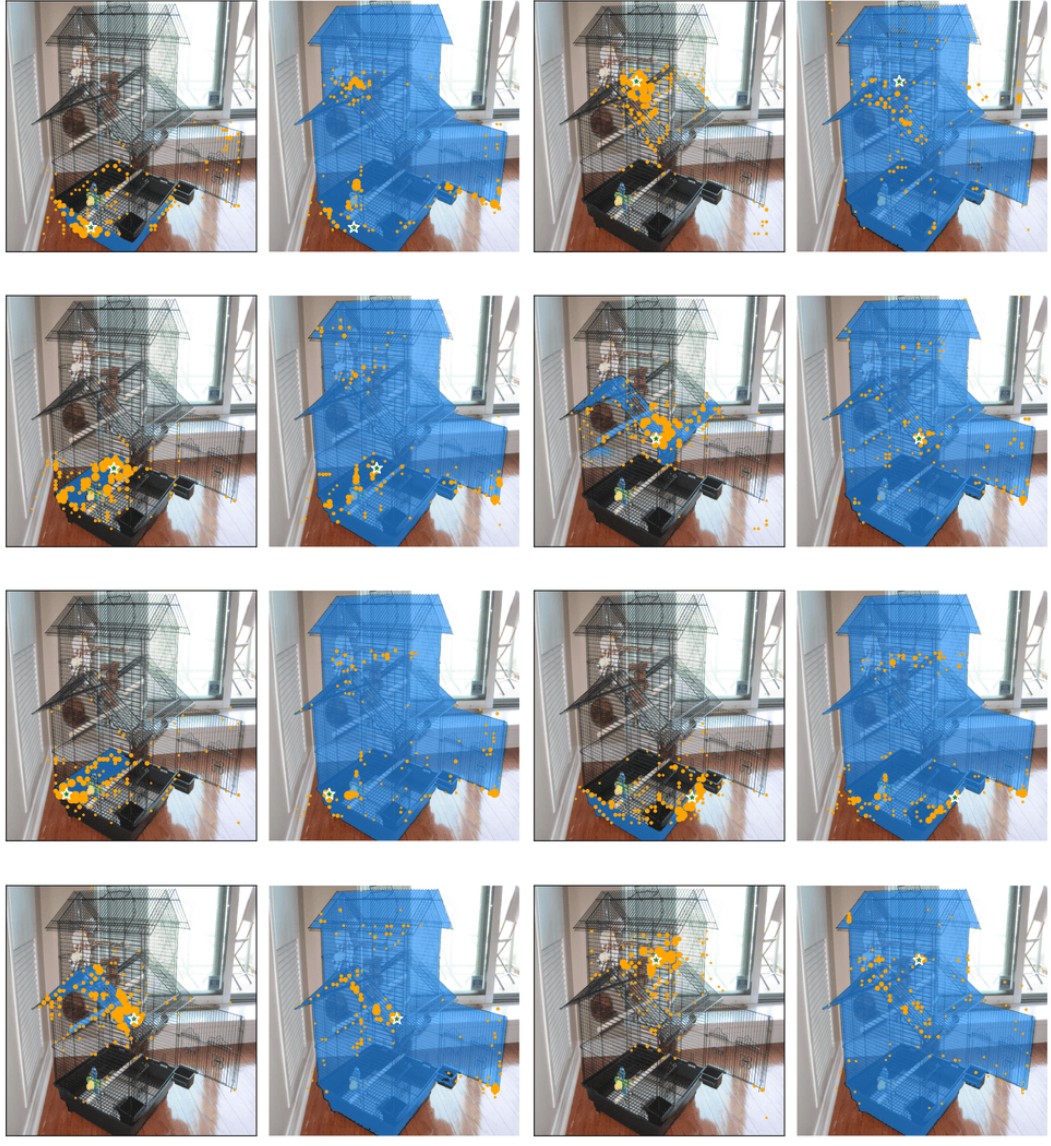

Figure 7: Visual results for 1-point prompt.

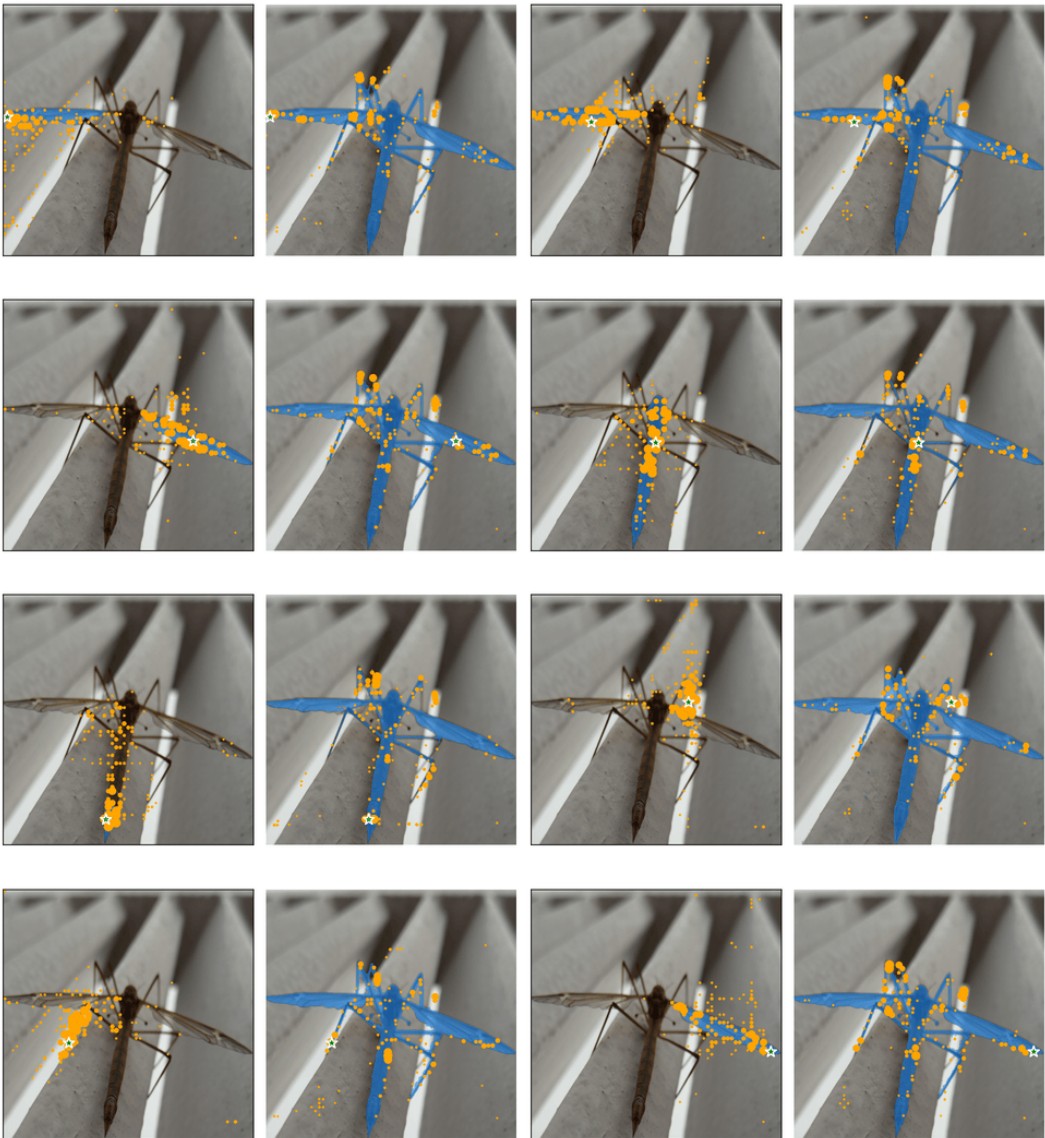

Figure 8: Visual results for 1-point prompt.

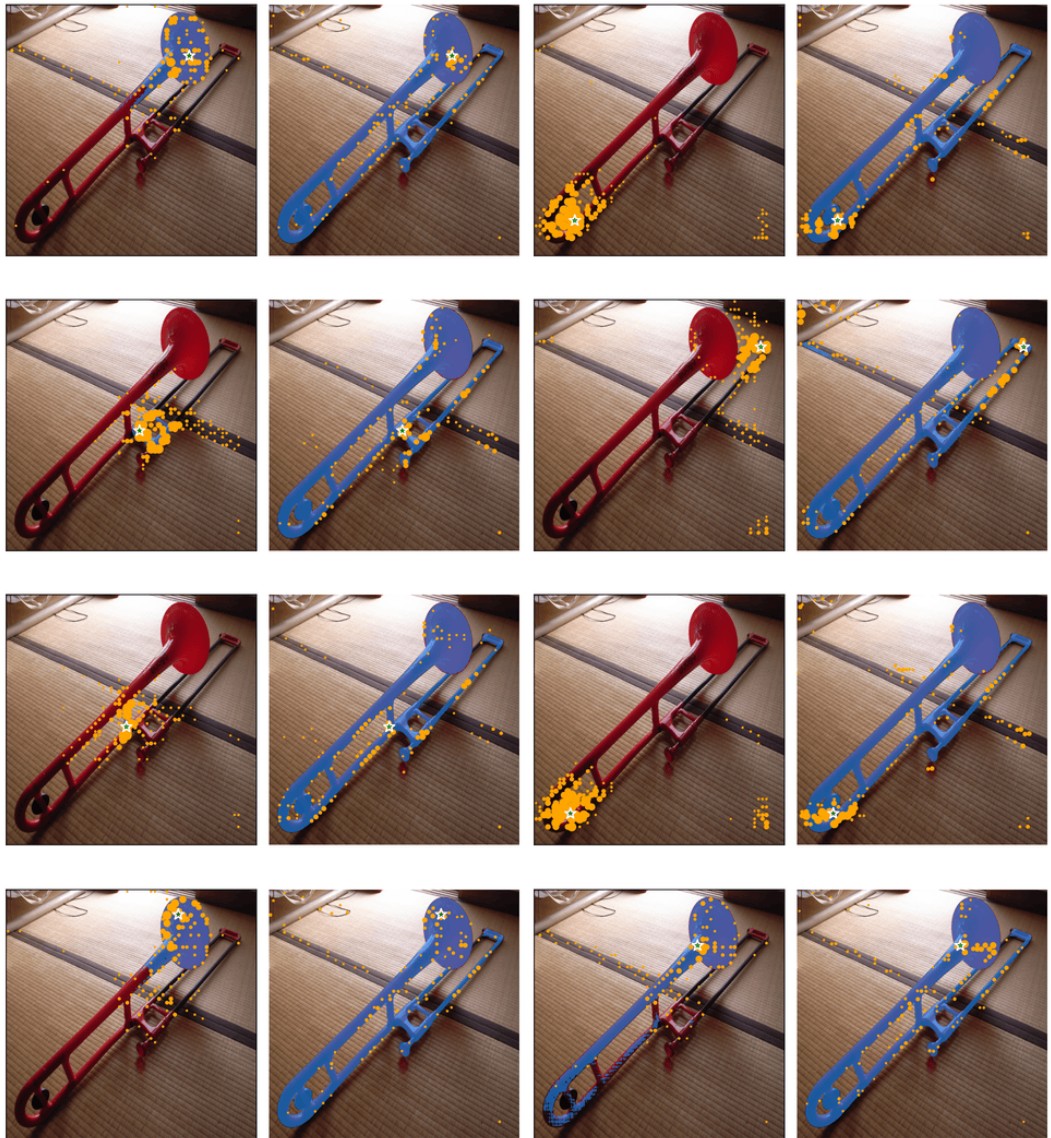

Figure 9: Visual results for 1-point prompt.

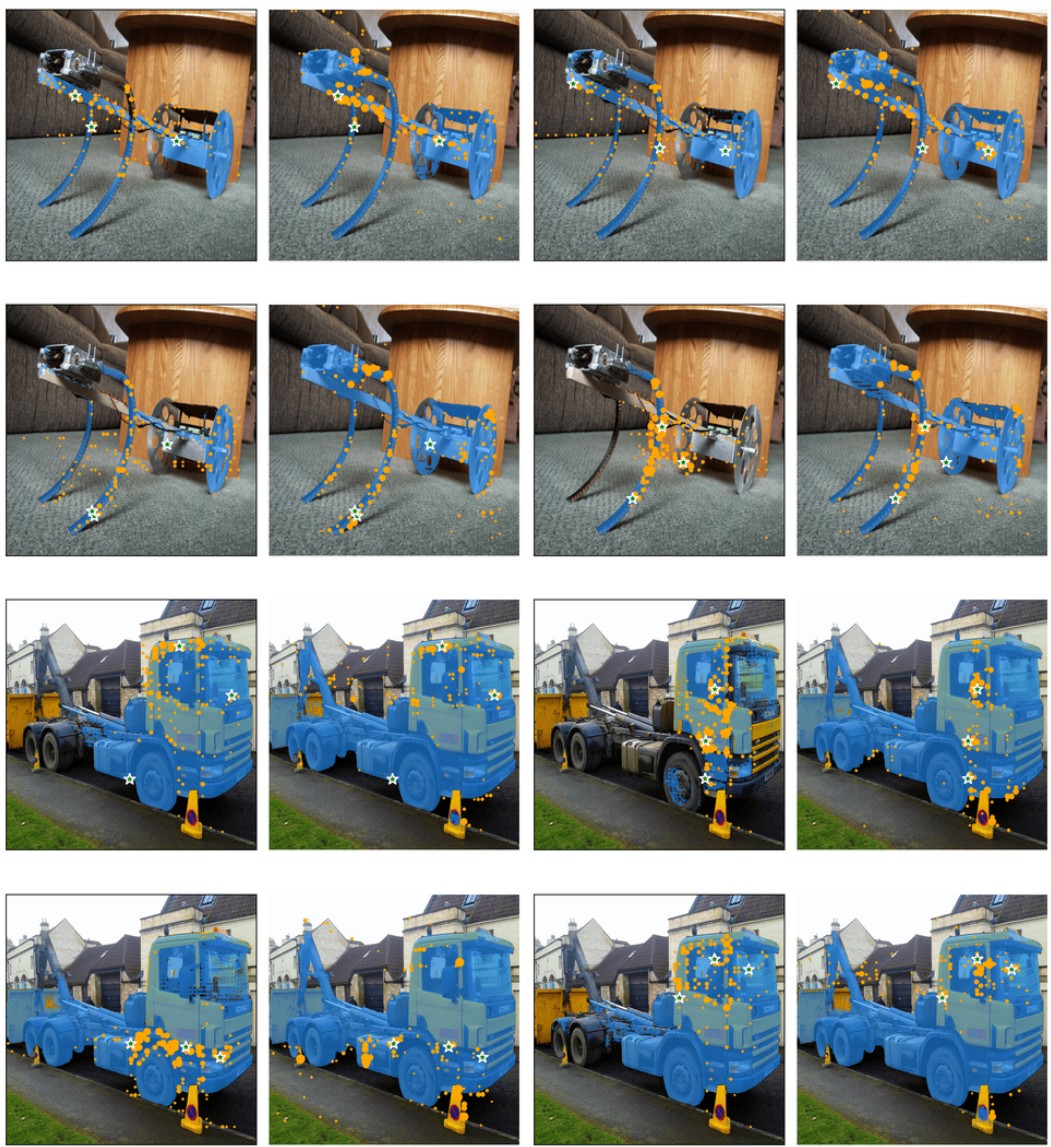

Figure 10: Visual results for 3-points prompt.

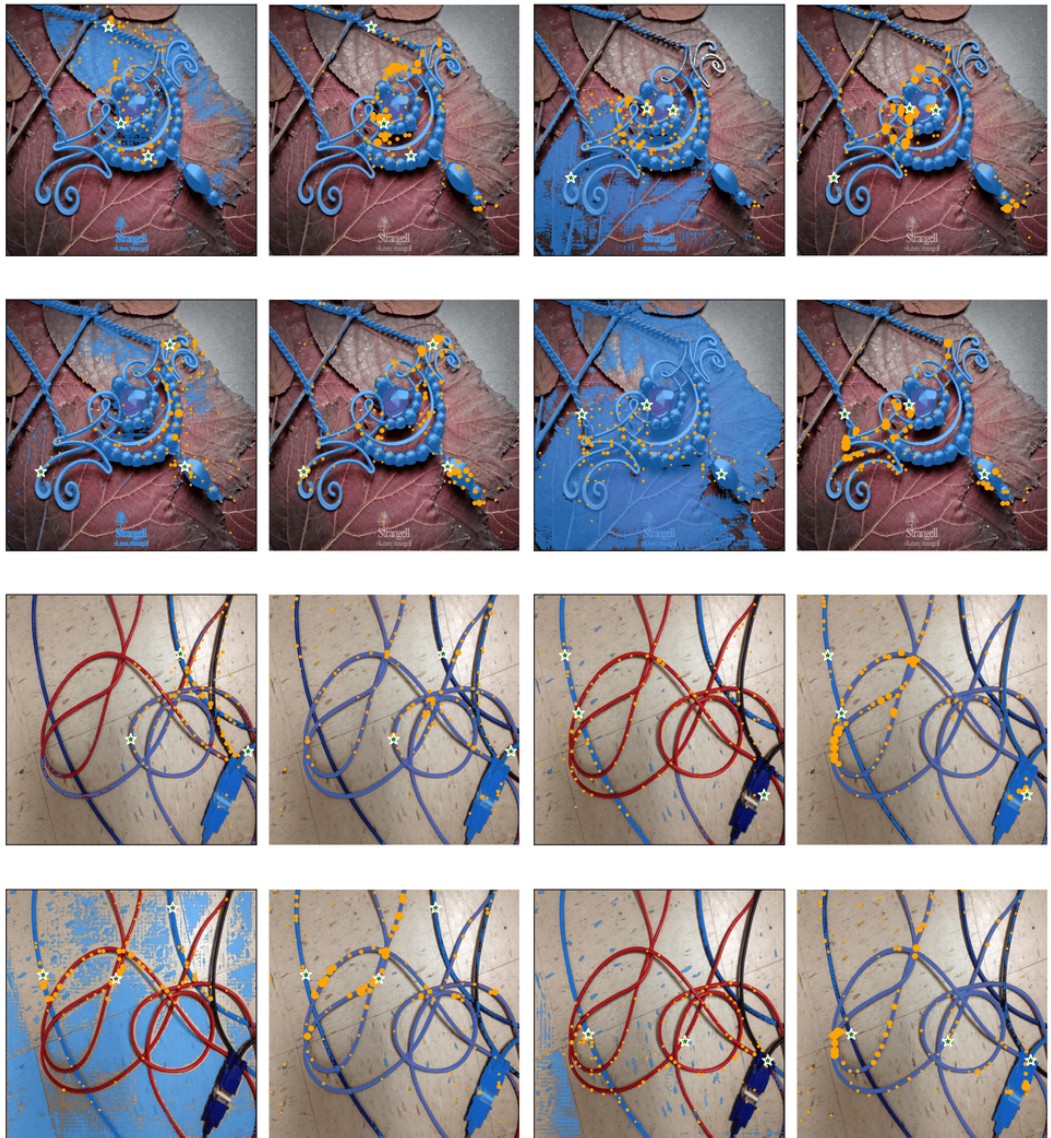

Figure 11: Visual results for 3-points prompt.

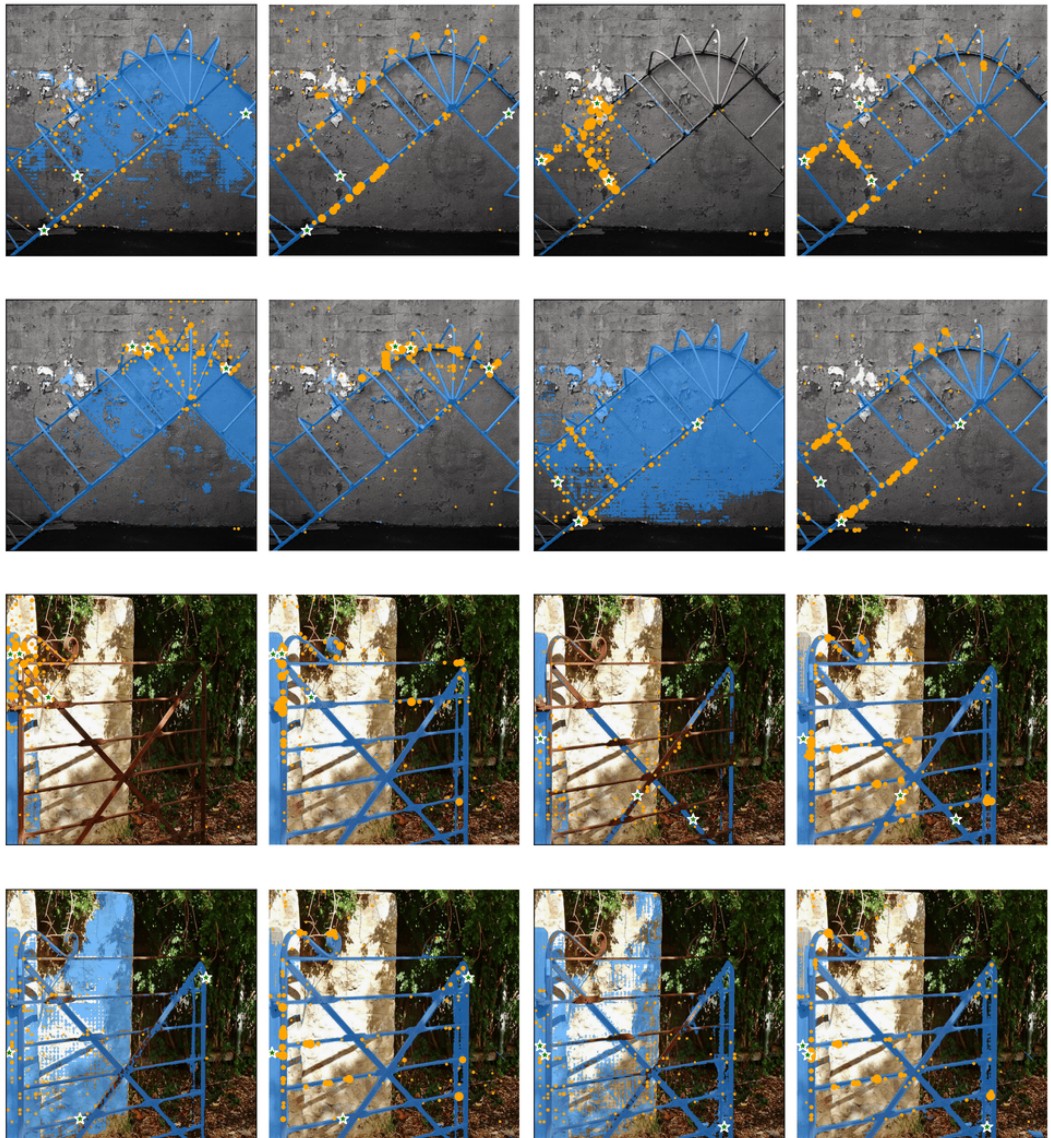

Figure 12: Visual results for 3-points prompt.

ACKNOWLEDGEMENTS

This research was supported in part by the Natural Science Foundation of Jiangsu Province (BK20241200), National Natural Science Foundation of China (62406140), Research Grant Council of the HKSAR (16201420), and Kuaishou Technology.