# OpenReview forum: "Stable Segment Anything Model"
_ICLR.cc/2025/Conference — ICLR 2025 Poster_

### Official Review · Reviewer_wxwP · 2024-10-21

**Soundness:** 3
**Presentation:** 3
**Contribution:** 3
**Rating:** 6
**Confidence:** 5

**Summary:**

This paper proposes Stable SAM, which aims to improve the segmentation ability of SAM under low-quality prompts. It focuses on the segmentation stability of SAM under different quality prompts and proposes a stability metric called ST. To improve the stability, Stable SAM proposes deformable sampling plugin (DSP) and dynamic routing plugin (DRP), which enable SAM to adaptively shift attention to the prompted target regions and switch SAM between the deformable and regular grid sampling modes.

**Strengths:**

1. The paper is well-written and well-presented.
2. The problem addressed in this paper is novel and focuses on the segmentation capabilities of SAM under low-quality prompts.
3. The results of this paper are good and the experiments are sufficient.

**Weaknesses:**

1. Why the adjustment of the attention sampling position can achieve calibration of noisy prompts through learnable offsets? This lacks intuitive motivation and theoretical justification. The visualization comparison in the paper does proves that this method can produce good generalization effects, but the transition here is a bit abrupt. I hope the author can explain the motivation behind this idea in more detail.

2. Line 245 of the paper mentions "Note that conventional deformable attention optimizes both the offset network and attention module. Thus directly applying deformable attention to SAM is usually suboptimal, because altering SAM’s original network or weights", but intuitively, retraining all components may be the optimal result. Freezing the parameters trained in a predetermined way and re-adding some parameters for adaptation seems to be a compromise solution and result. However, this is still a conjecture and there is no necessary evidence to prove this statement. Perhaps SAM needs to be pre-trained again.

3. The purpose of DRP is to control the DSP’s activation to prevent unwanted attention shifts. Are there any experiments that prove that the output of DRP is consistent with the expected intuition under different quality prompt inputs? In addition, in the case of precise prompt input, whether increasing the output strength of DRP leads to worse results, and vice versa. I think sufficient experiments are needed here to verify the intuitive conjecture of DRP.

4. RTS makes the model produce better performance, I think it is necessary to compare RTS+SAM (w/ and w/o deformable attention).

5. According to Table 7, I cannot understand why SAM+DSP can achieve a significant performance improvement (even in the case of a single click) with only a small amount of fine-tuning using high-quality prompts. I think this phenomenon is not in line with common sense and requires sufficient evidence for demonstration and analysis.

6. The strategy proposed in the "Robust Training Against Ambiguous Prompts" section will cause the model to tend to segment larger objects when faced with ambiguity, and cannot handle the possible multiple segmentation results expected by the user, which is not conducive to solving the ambiguity problem of interactive segmentation.

**Questions:**

Please refer to the Weaknesses.

---

> ### Author Response · Authors · 2024-11-23
> **Author Responses to Reviewer wxwP (Part 1/3)**
>
> ---
>
> We would like to sincerely thank you for your efforts and valuable comments to improve our work!
>
> Below we address your concerns.
>
> ---
>
> **Q1. Motivation behind DSP (Dynamic Sampling Plugin).**
>
> **A1.**
> Thanks for your valuable comment.
>
> **Our motivation has been elaborated in L187-L206, with both quantitative and qualitative analyses (Table 1, Figure 1, and Figure 2) in the main paper.**
> Here, we provide further explanation of our motivation.
>
> **Problem motivation:** The primary motivation for adjusting the attention sampling positions through learnable offsets is to overcome the inherent limitations of traditional fixed-grid attention, especially in the presence of noisy prompts.
> In the original SAM architecture, the mask decoder uses a regular grid sampling strategy, assuming that the initial prompt provides an accurate indication of the target region.
> However, in real-world scenarios where prompts are imprecise, fixed-grid sampling can cause the attention mechanism to focus on unintended areas, often capturing background or partially correct features.
>
> **Method motivation:** Our approach introduces learnable offsets to adapt the sampling positions dynamically, enabling attention to more precisely target the correct regions, even with ambiguous or noisy prompts.
> The learnable offsets enable the attention mechanism to "deform" the regions it focuses on in a data-driven manner, akin to how a person might adjust their focus when given unclear instructions.
> Essentially, the model learns from diverse training examples how to "interpret" and refine ambiguous or noisy prompts, using the learned offsets to direct attention to the most contextually relevant regions of the image.
>
>
> We have included this discussion in the revised supplementary material.
>
> ---
>
>
> **Q2. Parameter-efficient fine-tuning (PEFT) *v.s.* full retraining.**
>
> **A2.**
> Thank you for highlighting the trade-offs between retraining all components of SAM and adopting a parameter-efficient fine-tuning (PEFT) approach.
>
> **The decision to use PEFT rather than retraining all components of SAM primarily depends on the scale and type of the training dataset.**
> SAM was pretrained on the SA-1B dataset, containing 1 billion masks and 11 million images, which endows it with a generalized ability to segment a wide range of objects across various domains.
> In contrast, our downstream dataset, HQSeg-44K, contains only 44,000 images.
> Due to the significant difference in dataset scale, PEFT is a strategic choice to prevent overfitting while effectively adapting the pre-trained SAM to our target segmentation tasks with a small training budget.
>
>
> **Numerous studies have demonstrated that PEFT methods are particularly effective when adapting large pretrained models to downstream tasks with limited data.**
> This approach retains the generalizable features learned during pre-training, fine-tuning only a small subset of additional parameters.
> This paradigm is well-established in parameter-efficient fine-tuning research (as summarized in the PEFT surveys [1, 2]).
> By using PEFT, we aim to retain SAM's foundational representation power while mitigating overfitting, which is more likely to occur if we were to retrain the entire model on a relatively small dataset.
>
> In our case, finetuning only the DSP and DRP modules ensures that SAM retains its generality and adaptability while becoming more robust to noisy and ambiguous prompts, maintaining computational efficiency.
> In contrast, retraining SAM’s entire model compromises its integrity and significantly impairs performance, as shown by the large performance drop of FT-SAM (finetuning SAM’s whole model) **in Table 2 of the main paper**.
>
>
>
>
>
> We have included this discussion in the revised supplementary material.

---

> ### Author Response · Authors · 2024-11-23
> **Author Responses to Reviewer wxwP (Part 2/3)**
>
> ---
>
> **Q3. Analysis of DRP behavior under different prompt qualities.**
>
> **A3.**
> Thanks for your insightful comment.
>
>
> **We have conducted experiments to analyze the behavior of the DRP across different prompt qualities, as shown in Table 5 of the supplementary material.**
> We copy the table here for your reference.
>
> |      # of Points | 1 | 2 | 3 | 4 | 5 | 10 | 20 |
> | :---------------- | :------: | :----: | :----: | :----: | :----: | :----: | :----: |
> | α\_1 | 0.614 | 0.552 | 0.469 | 0.427 | 0.398 | 0.371 | 0.359 |
>
> The results show that the DSP routing weight, α\_1 (learned and controlled by DRP), decreases from 0.614 to 0.359 as the number of point prompts increases from one to twenty.
> This indicates that lower-quality prompts rely more on DSP features to direct attention to the desired regions.
>
> **Reviewer `UZpL` also appreciates that `The inclusion of a dynamic routing mechanism is practical and well-justified.`**
>
>
> Following your suggestion, we conduct additional experiments to manipulate the output strength of the DRP and examine its impact on segmentation quality for different prompt qualities.
> Specifically, we evaluate segmentation performance at different α\_1 values for both ambiguous (one point) and precise (ten points) prompts.
> For a fair comparison, we use the same set of point prompts for each α\_1 value.
>
> |   α\_1   | 0.0 | 0.2 |  0.4 | 0.5 | 0.6 | 0.8 | 1.0 |
> | :---------------- | :------: | :----: | :----: | :----: | :----: | :----: | :----: |
> | 1 point | 43.3 | 73.2 | 75.9 | 76.5 | 76.9 | 76.5 | 76.0 |
> | 10 points | 84.8 | 86.1 | 87.0 | 86.9 | 86.7 | 86.7 | 86.5 |
>
> The results show that larger α\_1 values (indicating stronger DSP activation) lead to better segmentation performance for the ambiguous one point prompt.
> This suggests that increasing the output strength of the DRP is particularly beneficial for less informative prompts.
> In contrast, segmentation performance is less sensitive to variations in α\_1 for precise prompts, suggesting that when prompts provide clearer guidance, the system achieves satisfactory results even with lower activation levels.
>
> We have included the discussion and experiments in the revised supplementary material.
>
> ---
>
> **Q4. Method comparisons with/without RTS.**
>
> **A4.**
> Thanks for your constructive suggestion.
>
> We have conducted experiments to analyze the behavior of the RTS for different methods, as shown **in Table 8 of the main paper**.
> We copy the table here for your reference.
>
> |       without RTS      | Groundtruth Box |  Noisy Box |
> | :---------------- | :------: | :----: |
> |  | mIoU & mBIoU | mIoU & mBIoU & ST |
> | SAM (baseline) | 79.5 & 71.1 | 48.8 & 42.1 & 39.5 |
> | PT-SAM | 87.6 & 79.7 | 70.6 & 60.4 & 64.0 |
> | HQ-SAM | 89.1 & 81.8 | 72.4 & 62.8 & 65.5 |
> | Ours (1 epoch) | 87.4 & 80.0 | 69.6 & 60.0 & 66.5 |
> | Ours (12 epochs) | 89.1 & 82.1 | 72.7 & 63.2 & 67.4 |
>
>
> |       with RTS      | Groundtruth Box |  Noisy Box |
> | :---------------- | :------: | :----: |
> |  | mIoU & mBIoU | mIoU & mBIoU & ST |
> | SAM (baseline) | 79.5 & 71.1 | 48.8 & 42.1 & 39.5 |
> | PT-SAM | 86.8 & 78.4 | 82.1 & 73.1 & 78.7 |
> | HQ-SAM | 87.4 & 79.8 | 82.9 & 74.5 & 80.4 |
> | Ours (1 epoch) | 86.0 & 78.4 | 82.3 & 74.1 & 82.3 |
> | Ours (12 epochs) | 87.4 & 80.1 | 84.4 & 76.7 & 85.2 |
> | HQ-SAM + Ours | 88.7 & 81.5 | 86.1 & 78.7 & 86.3 |
>
> Robust training is critical for improving model’s segmentation stability, but is
> usually overlooked in previous works. RTS can guide the model, including our DSP, to accurately segment target objects even when provided with misleading low-quality prompts. **The results show that RTS substantially improves the segmentation stability of all the methods**, albeit with a slight compromise in performance when dealing with high-quality prompts. **Note that our Stable-SAM benefits the most from the application of RTS, which can be attributed to our carefully designed deformable sampling plugin design.**

---

> ### Author Response · Authors · 2024-11-23
> **Author Responses to Reviewer wxwP (Part 3/3)**
>
> ---
>
> **Q5. Explaination on the performance improvement of SAM+DSP.**
>
> **A5.**
> Thanks for your helpful comment.
>
> In Table 7 of the main paper, SAM+DSP improves segmentation performance from 43.3 mIoU to 46.8 mIoU with the one-point prompt, where the 3.5 mIoU improvement is not as significant compared to the larger 33.6 mIoU improvement achieved by our full model.
>
> The large performance improvement with the noisy box prompt likely results from training on the HQSeg-44K dataset, a trend observed in other comparison methods, as shown in Table 1 of the main paper.
>
> The one-point prompt is more ambiguous than the noisy box prompt for indicating the desired segmentation areas.
> Thus, without RTS, the performance improvement with the one-point prompt is much smaller than with the noisy box prompt.
> Nevertheless, our DSP is effective in handling the challenge posed by the one-point prompt, outperforming other methods.
> DSP's flexibility enables the model to shift attention precisely to the intended region, compensating for the limited information provided by a single point.
> **Thus, DSP helps guide the attention mechanism toward a broader region that is contextually consistent with the high-quality prompt used during training.
> This adaptability can produce substantial gains even without RTS, which aligns with the observed results.**
>
>
> ---
>
> **Q6. Potential segmentation bias and ambiguity.**
>
> **A6.**
> Thank you for pointing out this interesting aspect.
>
> **We emphasize that our method is designed without inherent bias towards large or small objects.**
> Instead, the DSP adaptively redirects attention to align with the user’s intended target, whether a large or small object.
> Furthermore, the DRP toggles SAM between deformable and regular grid sampling modes to further mitigate concerns about potential bias.
> If the prompt explicitly targets small objects, our approach does not constrain the model to prioritize large objects, but instead adapts to the prompt’s ambiguity and specificity.
>
> While our approach does not introduce method bias, we acknowledge that the unbiased model may still be influenced by dataset bias.
> For example, if the dataset predominantly features large objects during training, the model may skew its predictions towards these large regions, potentially neglecting small objects.
> Thus, we also highlight the flexibility of our method.
> If users need to personalize their segmentation targets, such as focusing on specific small object regions, Stable-SAM can be easily fine-tuned to accommodate this requirement for better performance and stability.
> This fine-tuning process requires only a small amount of training data and minimal computational resources, as demonstrated in L512-L525 of the main paper.
> **This adaptability is a key strength, enabling our approach to effectively mitigate dataset bias and address a wide range of user needs and scenarios.**
>
> We have included this discussion in the revised supplementary material.
>
> ---
>
> **References**
>
> *[1] Parameter-Efficient Fine-Tuning for Large Models: A Comprehensive Survey, arXiv 2024*
>
> *[2] Parameter-Efficient Fine-Tuning for Pre-Trained Vision Models: A Survey, arXiv 2024*

---

> > ### Comment · Reviewer_wxwP · 2024-11-23
> >
> > Thanks to the author's response, most of my concerns have been addressed except for Q6.
> > Specifically, for the **occlusion image synthesis** of  more implementation details in the supplementary materials, whether this data augmentation method will lead to one-sided behavior of the model for ambiguous prompts. For example, when part of the quilt is on the bed and the other part is on the ground, and the user's requirement is only to segment the bed.

---

> > > ### Author Response · Authors · 2024-11-23
> > > **Author Responses to Reviewer wxwP**
> > >
> > > ---
> > >
> > > Thank you for your prompt engagement and insightful question.
> > >
> > > Our method does not lead to one-sided behavior in response to ambiguous prompts.
> > >
> > > **Our occlusion synthesis strategy is designed with diversity.** By randomly introducing occluders at varying scales and positions, we can train the model to handle situations where an object is partially visible, overlaps with another object, or remains unoccluded (occlusion synthesis is applied with a probability of 0.5 during training, as shown in L33 of the supplementary material).
> > > **This diversity prevents the model from developing biases that favor one interpretation over another in ambiguous scenarios.** Instead, the model learns to recognize consistent cues of the target object from the context and provided prompts.
> > >
> > >
> > > In your example, where part of a quilt is on a bed and another part is on the ground, our occlusion synthesis strategy enables that the model can differentiate between the bed and the quilt.
> > > **The model learns to segment the bed as a consistent target while recognizing that the quilt may or may not be part of the target, depending on context and prompts.**
> > > By consistently labeling objects in diverse contexts (*e.g.*, beds with and without quilts, occluded and unoccluded), our strategy enables the model to retain flexibility for accurate segmentation under varying conditions.
> > > **For example, if the box prompt encloses both the bed and the entire quilt, the model will likely segment both as a single entity. Otherwise, the model may segment only the bed.**
> > > This approach alleviates one-sided behavior, preventing incorrect segmentation of overlapping regions based solely on occlusion patterns.

---

> ### Comment · Reviewer_wxwP · 2024-11-25
>
> Thanks for the author's patient response. I understand the author's explanation, but I still retain some different opinions. I think that for the box prompt that encloses multiple objects, it is better to provide user multiple options. For example, for pillows/quilt that are all on the bed, users may still want to segment the bed separately, so providing multiple masks is always beneficial.
> I really appreciate the seriousness of the author. I decide to raise the score to 6.

---

> > ### Author Response · Authors · 2024-11-25
> > **Very happy that we addressed your concerns! Thanks for your valuable and insightful comments!**
> >
> > ---
> > **Thank you for your positive feedback. We are pleased to have addressed your concerns.**
> >
> > **We sincerely appreciate your valuable and insightful comments, and have greatly enjoyed this inspiring discussion.**
> >
> > We agree that both deterministic and non-deterministic segmentation are valuable for practical applications with varying purposes.
> > For instance, when annotating a dataset with well-defined custom segmentation targets (*e.g.*, e-commerce applications), Stable-SAM with deterministic output can enhance annotation efficiency.
> > On the other hand, for generic segmentation with diverse target intents, providing multiple masks with non-deterministic outputs is also beneficial.

---

### Official Review · Reviewer_Fktg · 2024-10-30

**Soundness:** 3
**Presentation:** 3
**Contribution:** 3
**Rating:** 6
**Confidence:** 4

**Summary:**

The paper presents Stable-SAM for enhancing SAM's segmentation stability when handling low-quality prompts. Stable-SAM employs DSP to adjust the sampling positions and amplitudes of image features without altering SAM's original architecture, which refines the mask attention. Additionally, the proposed DRP allows for switching between conventional and deformable sampling modes based on prompt quality, further improving the model's performance across varying prompt qualities. Experiments validate the method's effectiveness on multiple datasets.

**Strengths:**

1. This work provides new insights of SAM’s stability in segmentation and introduces a segmentation stability metric that effectively quantifies the stability.
2. The proposed DSP and DRP modules provide novel insights into the improvement of SAM.
3. Stable-SAM shows marked improvement with insufficient prompts.
4. Generally, the paper is easy to follow.

**Weaknesses:**

1. In Table 6, the interactive segmentation results for SAM on the SBD dataset appear inconsistent with those reported in other studies, showing higher performance. For example, in the works [1-4], the NoC90 metric for SAM-B/L/H exceeds 7.6, while Table 6 reports an SBD NoC90 of only 5.76. This discrepancy raises concerns given that this work builds upon SAM.

2. The core focus of this work is to address the issues arising from ambiguous prompts, yet SAM's practical application relies more on sufficiently detailed prompts, where SAM's inherent robustness may suffice. Moreover, Table 2 shows that increasing the number of points significantly reduces the advantage of Stable-SAM. Therefore, further experiments with more points are necessary to validate the effectiveness of Stable-SAM.

3. From another perspective, under precise prompts, the DSP may introduce shifts, as there remains a difference between DSP and human intent. If the intent is to segment small parts of the object, DSP may still guide the model to segment the main parts.

4. The paper lacks an analysis of computational efficiency. The DRP seems to run the SAM decoder twice (once with the new module proposed in the paper), leading to double the computational load of the SAM decoder. Thus, the efficiency analysis is necessary to confirm that the core performance benefits come from the new modules and not just increased computational demands.

Reference
1. ClickAttention: Click Region Similarity Guided Interactive Segmentation
2. FocSAM: Delving Deeply into Focused Objects in Segmenting Anything
3. MST: Adaptive Multi-Scale Tokens Guided Interactive Segmentation
4. PiClick: Picking the desired mask from multiple candidates in click-based interactive segmentation

**Questions:**

1. Please clarify the results for SAM in the SBD tests presented in Table 6.
2. How does Stable-SAM perform compared to other models with sufficient points?
3. Could you provide an analysis of the computational efficiency of Stable-SAM?

---

> ### Author Response · Authors · 2024-11-23
> **Author Responses to Reviewer Fktg (Part 1/2)**
>
> ---
>
> We would like to sincerely thank you for your efforts and valuable comments to improve our work!
>
> Below we address your concerns.
>
> ---
>
>
> **Q1. Clarification on the results for SAM in the SBD tests of Table 6.**
>
> **A1.**
> Thank you for your careful attention to the reported metrics in Table 6.
>
> The performance inconsistency can primarily be attributed to differences in experimental settings.
>
> In our experiments, we set `multimask_output = False`, while other methods likely evaluate SAM with the default `multimask_output = True` setting.
>
>
> As noted in SAM's original code, setting  `multimask_output = False` can yield better results for non-ambiguous prompts, such as multiple input prompts, as shown in `https://github.com/facebookresearch/segment-anything/blob/dca509fe793f601edb92606367a655c15ac00fdf/segment_anything/predictor.py#L115C1-L122C38`:
>
> ```
> multimask_output (bool): If true, the model will return three masks. For ambiguous input prompts (such as a single click), this will often produce better masks than a single prediction. If only a single mask is needed, the model's predicted quality score can be used to select the best mask. For non-ambiguous prompts, such as multiple input prompts, multimask_output=False can give better results.
> ```
>
> For a fair comparison, we reset `multimask_output = True`, retrained our method, and reevaluated the performance of both SAM and our method.
>
> |              | SBD | DAVIS |
> | :---------------- | :------: | :----: |
> |  | NoC85/NoC90 | NoC85/NoC90 |
> | FocalClick | 4.30/6.52 | 4.92/6.48 |
> | SimpleClick | 2.69/4.46 | 4.12/5.39|
> | SAM (ViT-L) | 5.93/7.51 | 4.78/5.96 |
> | PT-SAM (ViT-L)  | 4.03/5.40  | 4.27/5.42 |
> | Stable-SAM (ViT-L) | 2.93/4.59 | 4.01/5.13 |
>
>
>
> The results show that, with `multimask_output = True` setting, our reproduced SAM yields results consistent with those reported by other methods.
> Our method still performs better or comparably to other methods.
> We have updated Table 6 in the revised main paper.
>
>
> ---
>
>
> **Q2. Effectiveness of Stable-SAM with increasing number of points.**
>
> **A2.**
> Thank you for your valuable comment.
>
> We have already conducted ablation studies on the number of prompt points, as presented **in Table 4 of the supplementary material**.
> We copy Table 4 here for your reference.
>
> |      # of Points        | 1 | 3 | 5 | 7 | 10 | 15 | 20 |
> | :---------------- | :------: | :----: | :----: | :----: | :----: | :----: | :----: |
> | SAM | 43.3±1.8 | 78.7±0.8 | 83.3±0.7 | 84.1±0.7 | 84.8±0.6 | 85.3±0.6 | 85.7±0.5 |
> | PT-SAM | 43.0±1.9 | 80.1±0.7 | 84.3±0.6 | 85.2±0.6 | 85.6±0.4 | 86.1±0.4 | 86.3±0.3 |
> | Stable-SAM | 76.9±0.8 | 84.0±0.6 | 85.8±0.5 | 86.5±0.4 | 87.0±0.4 | 87.5±0.3 | 87.8±0.3 |
>
>
> The results show the performance curves of SAM, PT-SAM, and Stable-SAM for varying numbers of prompt points.
> We also report the performance standard deviation to assess segmentation stability.
>
> The results demonstrate that Stable-SAM achieves larger performance gains when handling lower-quality prompts, *i.e.*, fewer prompt points.
> As the number of prompt points increases, all methods begin to converge in performance, suggesting that while more prompts generally improve segmentation, the improvements become marginal beyond a certain number, *e.g.*, 7 points in this case.
> However, our method consistently outperforms others across a wide range of prompt point numbers, highlighting the robustness and adaptability of Stable-SAM in diverse prompting scenarios.

---

> ### Author Response · Authors · 2024-11-23
> **Author Responses to Reviewer Fktg (Part 2/2)**
>
> ---
>
> **Q3. Potential segmentation bias when targeting the small objects.**
>
>
> **A3.**
> Thank you for pointing out this interesting aspect.
>
> **We emphasize that our method is designed without inherent bias towards large or small objects.**
> Instead, the DSP adaptively redirects attention to align with the user’s intended target, whether a large or small object.
> Furthermore, the DRP toggles SAM between deformable and regular grid sampling modes to further mitigate concerns about potential bias.
> If the prompt explicitly targets small objects, our approach does not constrain the model to prioritize large objects, but instead adapts to the prompt’s ambiguity and specificity.
>
> While our approach does not introduce method bias, we acknowledge that the unbiased model may still be influenced by dataset bias.
> For example, if the dataset predominantly features large objects during training, the model may skew its predictions towards these large regions, potentially neglecting small objects.
> Thus, we also highlight the flexibility of our method.
> If users need to personalize their segmentation targets, such as focusing on specific small object regions, Stable-SAM can be easily fine-tuned to accommodate this requirement for better performance and stability.
> This fine-tuning process requires only a small amount of training data and minimal computational resources, as demonstrated in L512-L525 of the main paper.
> **This adaptability is a key strength, enabling our approach to effectively mitigate dataset bias and address a wide range of user needs and scenarios.**
>
>
> We have included this discussion in the revised supplementary material.
>
> ---
>
>
> **Q4. Computational efficiency analysis.**
>
>
> **A4.**
> Thank you for your constructive suggestion.
>
> **We clarify that our method runs the SAM decoder only once, with the DSP and DRP directly integrated into it.**
> This design ensures minimal computational overhead, as the new modules require only a small amount of additional processing time.
>
>
> We have already conducted a speed comparison (evaluated on the NVIDIA GeForce RTX 3090 GPU) **in Table 3 of the main paper**.
> We also include additional comparisons on model performance, parameters, inference speed, and GPU memory usage for different backbone variants **in Table 3 of the supplementary material**. We copy these tables here for your reference.
>
>
> |              | Learnable Params | FPS |
> | :---------------- | :------: | :----: |
> | SAM (baseline)    | (1191M) | 5.0 |
> | DT-SAM            | 3.9 M | 5.0 |
> | PT-SAM            | 0.13 M | 5.0 |
> | HQ-SAM            | 5.1 M | 4.8 |
> | Ours              | 0.08 M | 5.0 |
>
>
> |              | Total Params | Trainable Params | Memory | FPS |
> | :---------------- | :------: |:------: | :------: |  :----: |
> | SAM-Huge    | 2446 M | 2446 M | 10.3 G | 3.5 |
> | HQ-SAM-Huge           | 2452.1 M | 6.1 M | 10.3 G | 3.4 |
> | Stable-SAM-Huge            | 2446.08 M | 0.08 M | 10.3 G | 3.5 |
>
> |              | Total Params | Trainable Params | Memory | FPS |
> | :---------------- | :------: |:------: | :------: |  :----: |
> | SAM-Large    | 1191 M | 1191 M | 7.6 G | 5.0 |
> | HQ-SAM-Large           | 1196.1  M | 5.1 M | 7.6 G | 4.8 |
> | Stable-SAM-Large            | 1191.08 M | 0.08 M | 7.6 G | 5.0 |
>
> |              | Total Params | Trainable Params | Memory | FPS |
> | :---------------- | :------: |:------: | :------: |  :----: |
> | SAM-Base    | 358 M | 358 M |  5.1 G | 10.1 |
> | HQ-SAM-Base           | 362.1 M | 4.1 M | 5.1 G | 9.8 |
> | Stable-SAM-Base            | 358.08 M | 0.08 M | 5.1 G | 10.1 |
>
>
> The results demonstrate that our approach is lightweight and efficient, with the negligible addition of 0.08M parameters having no impact on the efficiency of the original SAM model.
>
> We have emphasized the efficiency advantages of our method in the revised main paper.

---

> > ### Comment · Reviewer_Fktg · 2024-11-25
> >
> > Thank you for addressing my concerns, especially regarding the replication of SAM results. Consequently, I increase the score. I recommend incorporating these replication details into the paper to inform related research efforts.

---

> > > ### Author Response · Authors · 2024-11-25
> > > **Very happy that we addressed your concerns! Thanks for your valuable and insightful comments!**
> > >
> > > ---
> > > **Thanks for your positive feedback!**
> > >
> > > **We are delighted that we addressed your concerns.**
> > >
> > > **We also sincerely appreciate your efforts to improve our work.**
> > >
> > > **We have adjusted our experimental setting to align with other methods.**
> > >
> > > **We have also added these implementation details in the revised supplementary material.**

---

### Official Review · Reviewer_UZpL · 2024-11-02

**Soundness:** 3
**Presentation:** 3
**Contribution:** 3
**Rating:** 8
**Confidence:** 4

**Summary:**

This paper introduces Stable-SAM, a variant of the Segment Anything Model (SAM), aimed at addressing SAM's instability when given low-quality prompts—either biased towards the background or confined to specific object parts. The authors propose two key modules integrated into SAM's mask decoder transformer: a Deformable Sampling Plugin (DSP) and a Deformable Routing Plugin (DRP). DSP adjusts the feature sampling offsets and amplitude, while DRP resamples deformable image features at updated sampling locations, refining SAM's token-to-image attention. Additionally, a dynamic routing mechanism is introduced to selectively activate DSP, preventing unwanted attention shifts. A robust training strategy is also proposed to help the model correct SAM's attention when negatively impacted by poor-quality prompts. Low-quality prompts are generated on fine-grained segmentation datasets, including DIS, ThinObject-5K, COIFT and HR-SOD, and the approach demonstrates clear performance improvements, outperforming SAM, SAM fine-tuned with LoRA, SAM fine-tuned with an adapter, and HQ-SAM.

**Strengths:**

1. The writing is clear, making the paper easy to follow.
2. The motivation to develop stable prompting for SAM is well-articulated and addresses a less-rexplored but valuable area. The empirical studies in Section 3 are solid and clearly presented.
3. The proposed DSP and DRP modules are novel, effectively enhancing SAM's stability without compromising its powerful pre-trained representations.The inclusion of a dynamic routing mechanism is practical and well-justified.
4. Stable-SAM demonstrates clear performance improvements over SAM and its listed variants.

**Weaknesses:**

1. The range of baseline models is limited. While many SAM-based models have recently been proposed to enhance its performance across various domains, this paper does not include comparisons with these relevant models.
2. The implementation of LoRA and adapters in SAM is not detailed. Since implementation choices can significantly affect performance, providing this information is crucial for evaluating the performance comparison as well as transparency and reproducibility.
3. The paper does not review SAM 2, which has been released and open-sourced. It would be nice to assess whether Stable-SAM is compatible with SAM 2 and if similar performance improvements can be achieved.

**Questions:**

1. While the authors have specified the number of trainable parameters, could they also provide the inference time of Stable-SAM compared to other baselines? I believe this is another important dimension to examine the efficiency.

2. (Minor) There is a typo: a period is missing at the end of Line 259.

---

> ### Author Response · Authors · 2024-11-23
> **Author Responses to Reviewer UZpL (Part 1/3)**
>
> ---
>
> We would like to sincerely thank you for your efforts and valuable comments to improve our work!
>
> Below we address your concerns.
>
> ---
>
>
> **Q1. Comparisons with more relevant SAM-based methods.**
>
> **A1.**
> Thank you for your valuable suggestion.
>
> In our initial submission, we focused the comparison on the original Segment Anything Model (SAM) as the primary baseline to isolate and highlight the effectiveness of our method in a controlled, well-understood environment.
>
> Following your suggestion, we compare our method with several recent SAM-based approaches with open-source code repositories.
> For a fair comparison, we train and evaluate these methods using the same datasets and experimental settings on four fine-grained segmentation datasets, as in the main paper.
>
>
> |              | Noisy Box | 1 Point | 3 Points |
> | :---------------- | :------: |:------: | :----: |
> | | mIoU & mBIoU & ST | mIoU & mBIoU & ST | mIoU & mBIoU & ST |
> | SAM (baseline)          |  48.8 & 42.1 & 39.5 | 43.3 & 37.4 & 45.1 | 78.7 & 69.5 & 79.3 |
> | PA-SAM [1]            | 51.2 & 44.4 & 41.8 | 45.3 & 39.5 & 47.2 | 79.6 & 70.1 & 80.0 |
> | CAT-SAM [2]           | 51.5 & 44.8 & 42.1 | 45.7 & 39.9 & 47.6 | 80.0 & 70.6 & 80.5 |
> | RobustSAM [3]         | 51.7 & 44.9 & 42.3 | 45.9 & 40.2 & 47.7 | 80.4 & 71.1 & 81.0 |
> | SAM 2 [4]             | 52.4 & 45.3 & 43.1 | 46.7 & 41.1 & 48.5 | 81.1 & 71.8 & 81.7 |
> | Ours                    | 82.3 & 74.1 & 82.3 | 76.9 & 68.4 & 71.1 | 84.0 & 75.8 & 84.9 |
>
> The results show that advanced SAM-based methods still suffer significantly from low-quality prompts.
> **This comparison not only highlights the strengths of our approach but also underscores the largely overlooked stability challenges inherent in SAM-based methods under varying prompt qualities.**
>
> We have included these experiments in the revised main paper.
>
>
>
>
>
>
> ---
>
>
>
>
>
> **Q2. More implementation details of LoRA and adapters in SAM.**
>
> **A2.**
> Thank you for your constructive feedback.
>
> We introduce Adapter/LoRA modules to the feed-forward network (FFN) of each ViT layer in SAM’s encoder for tuning.
> During training, we fine-tune only the adapter/LoRA modules and SAM’s prediction layer, with all other parameters frozen.
> In line with AdaptFormer [5], the adapters consist of small bottleneck layers inserted  in parallel  into the FFN, containing two MLPs and a GELU activation function between them.
> The bottleneck’s middle dimension is set to 64 to balance model performance and computational efficiency.
> In line with LoRA [6], the module uses an encoder-decoder structure to impose a low-rank constraint on FFN weight updates, injecting small trainable rank decomposition matrices into each layer.
> In our experiments, the rank of LoRA is set to 4 for efficiency and performance optimization.
> All other experimental settings remain the same as those of the baseline and full model.
>
> We have included these implementation details in the revised supplementary material.

---

> ### Author Response · Authors · 2024-11-23
> **Author Responses to Reviewer UZpL (Part 2/3)**
>
> ---
>
> **Q3. Review, Comparison, and Extension to SAM 2.**
>
> **A3.**
> Thanks for your insightful suggestion.
>
> Segment Anything Model 2 (SAM 2) is a unified model for video and image-based promptable segmentation.
> SAM 2 is a natural extension of SAM to the video domain, incorporating a memory attention module that leverages object and interaction information from previously observed frames.
> When applied to images, SAM 2 behaves similarly to SAM, with the memory module empty.
> SAM 2 also collects the largest video segmentation dataset for training.
> Therefore, for image-based promptable segmentation, SAM 2 benefits from a larger training dataset, a stronger hierarchical backbone [7, 8], and multi-scale feature decoding in the mask decoder.
>
> **Our method can be seamlessly integrated into SAM 2 to enhance segmentation stability and performance under prompts of varying quality.**
> We train the original SAM 2 model with the Hiera-B+ backbone on the HQSeg-44K dataset and evaluate its performance on four fine-grained segmentation datasets, using the same experimental settings as in our main experiments.
> We further apply our Stable-SAM method to SAM 2, resulting in the Stable-SAM 2 variant.
> We compare Stable-SAM 2 with SAM 2 and other state-of-the-art SAM-based methods in the table below.
>
> |              | Noisy Box | 1 Point | 3 Points |
> | :---------------- | :------: |:------: | :----: |
> | | mIoU & mBIoU & ST | mIoU & mBIoU & ST | mIoU & mBIoU & ST |
> | SAM (baseline)          |  48.8 & 42.1 & 39.5 | 43.3 & 37.4 & 45.1 | 78.7 & 69.5 & 79.3 |
> | SAM 2 [4]             | 52.4 & 45.3 & 43.1 | 46.7 & 41.1 & 48.5 | 81.1 & 71.8 & 81.7 |
> | Stable-SAM                    | 82.3 & 74.1 & 82.3 | 76.9 & 68.4 & 71.1 | 84.0 & 75.8 & 84.9 |
> | Stable-SAM 2           | 83.5 & 75.3 & 83.4 | 78.0 & 69.6 & 72.2 | 85.1 & 76.9 & 86.0 |
>
>
>
> SAM 2 outperforms SAM, due to its stronger backbone and larger pretraining dataset.
> However, SAM 2 still suffers significantly from low-quality prompts, owing to the overlooked segmentation stability problem.
> Stable-SAM 2 exhibits substantial improvements in segmentation quality and stability, outperforming the original Stable-SAM.
> **This confirms the effectiveness and applicability of our method in addressing segmentation stability across different SAM-based methods.**
>
>
> We have included these discussions and experiments in the revised main paper.
>
>
>
> ---
>
>
> **Q4. Inference Speed Comparisons.**
>
> **A4.**
> Thank you for your constructive suggestion.
>
> We have already conducted a speed comparison (evaluated on the NVIDIA GeForce RTX 3090 GPU) **in Table 3 of the main paper**.
> We also include additional comparisons on model performance, parameters, inference speed, and GPU memory usage for different backbone variants **in Table 3 of the supplementary material**.
> We copy those results here for your reference.
>
>
>
> |              | Learnable Params | FPS |
> | :---------------- | :------: | :----: |
> | SAM (baseline)    | (1191M) | 5.0 |
> | DT-SAM            | 3.9 M | 5.0 |
> | PT-SAM            | 0.13 M | 5.0 |
> | HQ-SAM            | 5.1 M | 4.8 |
> | Ours              | 0.08 M | 5.0 |
>
>
> |              | Total Params | Trainable Params | Memory | FPS |
> | :---------------- | :------: |:------: | :------: |  :----: |
> | SAM-Huge    | 2446 M | 2446 M | 10.3 G | 3.5 |
> | HQ-SAM-Huge           | 2452.1 M | 6.1 M | 10.3 G | 3.4 |
> | Stable-SAM-Huge            | 2446.08 M | 0.08 M | 10.3 G | 3.5 |
>
> |              | Total Params | Trainable Params | Memory | FPS |
> | :---------------- | :------: |:------: | :------: |  :----: |
> | SAM-Large    | 1191 M | 1191 M | 7.6 G | 5.0 |
> | HQ-SAM-Large           | 1196.1  M | 5.1 M | 7.6 G | 4.8 |
> | Stable-SAM-Large            | 1191.08 M | 0.08 M | 7.6 G | 5.0 |
>
> |              | Total Params | Trainable Params | Memory | FPS |
> | :---------------- | :------: |:------: | :------: |  :----: |
> | SAM-Base    | 358 M | 358 M |  5.1 G | 10.1 |
> | HQ-SAM-Base           | 362.1 M | 4.1 M | 5.1 G | 9.8 |
> | Stable-SAM-Base            | 358.08 M | 0.08 M | 5.1 G | 10.1 |
>
>
> The results demonstrate that our approach is lightweight and efficient, with the negligible addition of 0.08M parameters having no impact on the efficiency of the original SAM model.
> We will emphasize the efficiency advantages of our method in the main paper.
>
> We have emphasized the efficiency advantages of our method in the revised main paper.

---

> ### Author Response · Authors · 2024-11-23
> **Author Responses to Reviewer UZpL (Part 3/3)**
>
> ---
>
> **Q5. Writing typo.**
>
> **A5.**
> Thank you for your helpful comment.
>
> We have already corrected this typo and will conduct further proofreading to improve the overall quality of the manuscript.
>
> ---
>
> **References**
>
> *[1] PA-SAM: Prompt Adapter SAM for High-quality Image Segmentation, ICME 2024*
>
> *[2] CAT-SAM: Conditional Tuning for Few-Shot Adaptation of Segment Anything Model, ECCV 2024*
>
> *[3] RobustSAM: Segment Anything Robustly on Degraded Images, CVPR 2024*
>
> *[4] SAM 2: Segment Anything in Images and Videos, arXiv 2024*
>
> *[5] AdaptFormer: Adapting Vision Transformers for Scalable Visual Recognition, NeurIPS 2022*
>
> *[6] LoRA: Low-Rank Adaptation of Large Language Models, ICLR 2022*
>
> *[7] Hiera: A hierarchical vision transformer without the bells-and-whistles, ICML 2023*
>
> *[8] Window attention is bugged: How not to interpolate position embeddings, ICLR 2023*

---

> > ### Comment · Reviewer_UZpL · 2024-11-27
> >
> > Thanks for the detailed reply. The responses addressed most of my concerns and I will keep my score.

---

> > > ### Author Response · Authors · 2024-11-29
> > > **Very happy that we addressed your concerns! Thanks for your valuable and insightful comments!**
> > >
> > > ---
> > >
> > > **Thanks for your positive feedback!**
> > >
> > > **We are delighted that we addressed your concerns.**
> > >
> > > **We also sincerely appreciate your efforts to improve our work.**

---

### Official Review · Reviewer_SsTF · 2024-11-04

**Soundness:** 3
**Presentation:** 3
**Contribution:** 3
**Rating:** 6
**Confidence:** 4

**Summary:**

This paper analyzes the stability issue of segment anything models (SAMs). It is observed that SAMs tend to focus on background or some undesired parts when given low-quality prompts. A metric of segmentation stability is introduced to better evaluate this capability. Two main modules are proposed to improve the stability of SAMs, i.e. deformable sampling to organize the image feature and dynamic routing to allocate the magnitude of deformed activation.

**Strengths:**

1. The topic this paper focuses on is interesting and important, that may be ignored in previous research. Though SAMs produce surprising segmentation results, they may fail when the input prompts are not that accurate, especially in real-life applications.
2. The proposed method is simple but effective. The promotion over previous SAM and high-quality SAM methods is notable on several benchmarks and tasks.
3. The writing is easy to follow.

**Weaknesses:**

1. One potential negative effect to discuss. Though this paper targets strengthening the ability of SAM to focus on the foreground object, will it tend to produce wrong segmentation results when the desired segmenting part is the background? It will be interesting to discuss the bias this method may introduce.
2. It is easy to demonstrate the improvement when the number of prompts reduces to 1 or 3 points. However, noisy boxes are usually determined by the user, which are more subjective. Given a user has priors to roughly locate the desired object, the noise may not be that big. Therefore, a user study is recommended to better demonstrate how much benefit the method can actually bring when applied to real scenarios. The users can be required to give prompts to the same set of testing data, which can be more realistic than generated noisy prompts.
3. Though the proposed modules are light and parameters-friendly, it is also necessary to evaluate their real latency and compare them with other methods.

**Questions:**

One open question: the module is efficient in improving the stability, considering both the training and inference budget. However, will scaling up the training data with more noisy data also improve the segmentation stability? How to treat the two factors?

---

> ### Author Response · Authors · 2024-11-23
> **Author Responses to Reviewer SsTF (Part 1/2)**
>
> ---
>
> We would like to sincerely thank you for your efforts and valuable comments to improve our work!
>
> Below we address your concerns.
>
> ---
>
> **Q1. Potential segmentation bias when targeting the background.**
>
> **A1.** Thank you for pointing out this interesting aspect.
>
> **We emphasize that our method is designed without inherent bias towards large or small objects.**
> Instead, the DSP adaptively shifts attention to match the user’s intended target, whether a foreground object or background area.
> Additionally, the DRP toggles SAM between deformable and regular grid sampling modes to reduce concerns regarding potential bias.
> If the prompt explicitly specifies the background, our method adapts accordingly, without constraining the model to prioritize the foreground.
>
>
> Although our method avoids introducing bias, we acknowledge that the model may still be influenced by dataset bias.
> For instance, if the training dataset predominantly contains foreground objects, the model may skew predictions towards the foreground, potentially neglecting background regions.
> Thus, we also highlight the flexibility of our method.
> If users wish to personalize segmentation targets, such as focusing on specific background regions, Stable-SAM can be easily fine-tuned to meet this requirement.
> This fine-tuning process requires minimal training data and budget, as shown in L512-L525 of the main paper.
> **This adaptability is a key strength, enabling our approach to effectively mitigate dataset bias and address a wide range of user needs and scenarios beyond typical foreground segmentation.**
>
> We have included this discussion in the revised supplementary material.
>
>
> ---
>
> **Q2. User study of noisy boxes for realistic application scenarios.**
>
> **A2.** Thank you for your insightful comment.
>
> In response to your suggestion, we conduct a user study to provide a more realistic evaluation of our method, using noisy box annotated from users across four fine-grained segmentation datasets: DIS (validation set), ThinObject5K (test set), COIFT, and HR-SOD.
> Five participants are asked to provide box annotations for the highlighted target object in each image.
> Participants are instructed to complete each box annotation within 5 seconds to ensure consistency throughout the process.
> Annotations are collected using the Label Studio platform.
> The user-annotated boxes are subsequently used as prompts to evaluate the performance of each segmentation method.
>
>
> |                         | Noisy Box |  User-Annotated Box |
> | :---------------- | :------: | :----: |
> |  | mIoU & mBIoU & ST | mIoU & mBIoU & ST |
> | SAM (baseline)    | 48.8 & 42.1 & 39.5 | 58.3 & 51.7 & 49.3 |
> | HQ-SAM            | 72.4 & 62.8 & 65.5 | 76.1 & 66.4 & 69.2 |
> | Ours              | 82.3 & 74.1 & 82.3 | 84.2 & 76.3 & 84.0 |
>
> The results show that user-annotated boxes provide better segmentation performance across all methods compared to generated noisy boxes.
> We also assess the quality of the user-annotated boxes by comparing them to the ground truth boxes derived from the mask annotations.
> The average IoU between user-annotated boxes and ground truth boxes is approximately 0.753, indicating that user-annotated boxes are more accurate than generated noisy boxes.
> **Under the user-provided box prompt, our method continues to outperform other methods by a large margin.**
>
> We have included this experiment in the revised supplementary material.

---

> ### Author Response · Authors · 2024-11-23
> **Author Responses to Reviewer SsTF (Part 2/2)**
>
> ---
>
> **Q3. Inference Speed Comparisons.**
>
> **A3.**
> Thank you for your constructive suggestion.
>
> We have already conducted a speed comparison (evaluated on the NVIDIA GeForce RTX 3090 GPU) **in Table 3 of the main paper**.
> We have also included additional comparisons on model performance, parameters, inference speed, and GPU memory usage for different backbone variants **in Table 3 of the supplementary material**.
> We copy those results here for your reference.
>
>
> |              | Learnable Params | FPS |
> | :---------------- | :------: | :----: |
> | SAM (baseline)    | (1191M) | 5.0 |
> | DT-SAM            | 3.9 M | 5.0 |
> | PT-SAM            | 0.13 M | 5.0 |
> | HQ-SAM            | 5.1 M | 4.8 |
> | Ours              | 0.08 M | 5.0 |
>
>
> |              | Total Params | Trainable Params | Memory | FPS |
> | :---------------- | :------: |:------: | :------: |  :----: |
> | SAM-Huge    | 2446 M | 2446 M | 10.3 G | 3.5 |
> | HQ-SAM-Huge           | 2452.1 M | 6.1 M | 10.3 G | 3.4 |
> | Stable-SAM-Huge            | 2446.08 M | 0.08 M | 10.3 G | 3.5 |
>
> |              | Total Params | Trainable Params | Memory | FPS |
> | :---------------- | :------: |:------: | :------: |  :----: |
> | SAM-Large    | 1191 M | 1191 M | 7.6 G | 5.0 |
> | HQ-SAM-Large           | 1196.1  M | 5.1 M | 7.6 G | 4.8 |
> | Stable-SAM-Large            | 1191.08 M | 0.08 M | 7.6 G | 5.0 |
>
> |              | Total Params | Trainable Params | Memory | FPS |
> | :---------------- | :------: |:------: | :------: |  :----: |
> | SAM-Base    | 358 M | 358 M |  5.1 G | 10.1 |
> | HQ-SAM-Base           | 362.1 M | 4.1 M | 5.1 G | 9.8 |
> | Stable-SAM-Base            | 358.08 M | 0.08 M | 5.1 G | 10.1 |
>
>
>
> The results demonstrate that our approach is lightweight and efficient, with the negligible addition of 0.08M parameters having no impact on the efficiency of the original SAM model.
>
> We have emphasized the efficiency advantages of our method in the revised main paper.
>
> ---
>
> **Q4. Model scalability to noisy training data.**
>
>
> **A4.**
> Thank you for your insightful question.
>
> Increasing the amount of training data could improve the model’s robustness and generalization.
> More noisy data could help the model learn to handle a wider variety of input prompts, especially in real-world scenarios where user-provided prompts are often imprecise or ambiguous. As the model is exposed to more diverse and challenging inputs, it may become better at distinguishing relevant features and handling uncertainty in segmentation.
> However, introducing excessive noisy data could cause the model to overfit to the noise, leading to instability in some cases.
> **Our method, which dynamically calibrates attention based on prompt quality, mitigates some risks of noisy data by guiding the model to focus on relevant image regions without being overwhelmed by noise.**
>
>
> We also understand your concerns about the scalability of our method when scaling up the training data.
> While our method is designed to efficiently handle noisy data with limited resources, it can still benefit from larger training sets.
> **As the training dataset grows, we can fine-tune additional parameters of Stable-SAM to further improve segmentation stability.**
> Our method’s lightweight design, with only 0.08M learnable parameters, allows for fine-tuning additional parameters without significantly increasing model complexity or computational cost.
> This additional fine-tuning of more parameters applied to larger datasets, can further boost performance and stability, as the model adapts more precisely to the increased diversity of data and ambiguous prompts.
> **This highlights the flexibility of our approach, which performs well with limited data but can also benefit from additional fine-tuning of more parameters when the training dataset increases in size and noise.**
>
>
> We have included this discussion in the revised supplementary material.

---

### Author Response · Authors · 2024-11-23
**Author General Responses**

---

**We would like to sincerely thank all reviewers for your efforts and valuable comments to improve our work!**

---

**We have uploaded a new version of our main paper and supplementary material, revised based on reviewers’ valuable and helpful comments. We highlight the revised parts in blue color for better reference.**

---

**We are pleased that all reviewers consistently appreciate our work's novelty, significance, method effectiveness, extensive experiments, and writing quality.**

**1. Reviewers appreciate our significance, novelty and new insights:**

- The topic this paper focuses on is interesting and important, that may be ignored in previous research. (*Reviewer `SsTF`*)
- The motivation to develop stable prompting for SAM is well-articulated and addresses a less-rexplored but valuable area. (*Reviewer `UZpL`*)
- The proposed DSP and DRP modules are novel. (*Reviewer `UZpL`*)
- This work provides new insights of SAM’s stability in segmentation. (*Reviewer `Fktg`*)
- The proposed DSP and DRP modules provide novel insights into the improvement of SAM. (*Reviewer `Fktg`*)
- The problem addressed in this paper is novel. (*Reviewer `wxwP`*)


**2. Reviewers appreciate our method effectiveness and extensive experiments:**

- The proposed method is simple but effective. (*Reviewer `SsTF`*)
- The empirical studies in Section 3 are solid and clearly presented. (*Reviewer `UZpL`*)
- Stable-SAM demonstrates clear performance improvements over SAM and its listed variants. (*Reviewer `UZpL`*)
- Experiments validate the method's effectiveness on multiple datasets. (*Reviewer `Fktg`*)
- Stable-SAM shows marked improvement with insufficient prompts. (*Reviewer `Fktg`*)
- The results of this paper are good and the experiments are sufficient. (*Reviewer `wxwP`*)


 **3. Reviewers appreciate our good writing:**


- The writing is easy to follow. (*Reviewer `SsTF`*)
- The writing is clear, making the paper easy to follow. (*Reviewer `UZpL`*)
- Generally, the paper is easy to follow. (*Reviewer `Fktg`*)
- The paper is well-written and well-presented. (*Reviewer `wxwP`*)

---

### Meta-Review · Area_Chair_N3xr · 2024-12-18

**Metareview:**

The paper proposes a method to enhance the performance of SAM when provided with low-quality prompts through two novel modules: the Deformable Sampling Plugin (DSP) and the Dynamic Routing Plugin (DRP). Overall, the idea is novel and the results are solid. While the original submission lacked some details and experimental results, these shortcomings have been effectively addressed in the authors' feedback. Finally, all four reviewers provided positive scores. Based on the novelty of the approach, the solid experimental results, and the thoroughness of the authors' responses, the decision is to recommend this paper for acceptance.

**Additional Comments On Reviewer Discussion:**

The original submission lacked some details and experimental results. However, these concerns have been successfully addressed in the authors' feedback. Finally, all the 4 reviewers gave positive scores: 6, 6,6, and 8.

---

### Decision · Program_Chairs · 2025-01-22

Accept (Poster)